# Slo2 potassium channel function depends on RNA editing-regulated expression of a SCYL1 protein

**Long-Gang Niu, Ping Liu, Zhao-Wen Wang, Bojun Chen\***

Department of Neuroscience, University of Connecticut Health Center, Farmington, United States

**Abstract** Slo2 potassium channels play important roles in neuronal function, and their mutations in humans may cause epilepsies and cognitive defects. However, it is largely unknown how Slo2 is regulated by other proteins. Here we show that the function of *C. elegans* Slo2 (SLO-2) depends on *adr-1*, a gene important to RNA editing. ADR-1 promotes SLO-2 function not by editing the transcripts of *slo-2* but those of *scyl-1*, which encodes an orthologue of mammalian SCYL1. Transcripts of *scyl-1* are greatly decreased in *adr-1* mutants due to deficient RNA editing at a single adenosine in their 3'-UTR. SCYL-1 physically interacts with SLO-2 in neurons. Single-channel open probability ($P_o$) of neuronal SLO-2 is ~50% lower in *scyl-1* knockout mutant than wild type. Moreover, human Slo2.2/Slack $P_o$ is doubled by SCYL1 in a heterologous expression system. These results suggest that SCYL-1/SCYL1 is an evolutionarily conserved regulator of Slo2 channels.

## Introduction

Slo2 channels are large-conductance potassium channels existing in mammals as well as invertebrates (*Kaczmarek, 2013*; *Yuan et al., 2000*). They are the primary conductor of delayed outward currents in many neurons examined (*Budelli et al., 2009*; *Liu et al., 2014*). Human and mouse each has two Slo2 channels (Slo2.1/Slick and Slo2.2/Slack) (*Kaczmarek, 2013*), whereas the nematode *C. elegans* has only one (SLO-2). These channels are abundantly expressed in the nervous system (*Bhattacharjee et al., 2002*; *Bhattacharjee et al., 2005*; *Joiner et al., 1998*; *Liu et al., 2018*; *Rizzi et al., 2016*), and play major roles in shaping neuronal electrical properties and regulating neurotransmitter release (*Kaczmarek, 2013*; *Liu et al., 2014*). Mutations of Slo2 channels cause epilepsies and severe intellectual disabilities in humans (*Ambrosino et al., 2018*; *Cataldi et al., 2019*; *Evely et al., 2017*; *Gururaj et al., 2017*; *Hansen et al., 2017*; *Kawasaki et al., 2017*; *Lim et al., 2016*; *McTague et al., 2018*; *Rizzo et al., 2016*), and reduced tolerance to hypoxic environment in worms (*Yuan et al., 2003*). Emerging evidence suggests that physiological functions of these channels depend on other proteins. For example, in mice, the fragile mental retardation protein (FMRP), a RNA binding protein, enhances Slack activity by binding to its carboxyl terminus (*Brown et al., 2010*). In worms, HRPU-2, a RNA/DNA binding protein, controls the expression level of SLO-2 through a posttranscriptional effect (*Liu et al., 2018*).

RNA editing is an evolutionarily conserved post-transcriptional process catalyzed by ADARs (*adenosine deaminases acting on RNA*) (*Gott and Emeson, 2000*; *Jin et al., 2009*). ADARs convert adenosine (A) to inosine (I) in double-stranded RNA. Since inosine is interpreted as guanosine (G) by cellular machineries (*Basilio et al., 1962*), A-to-I RNA editing may alter the function of a protein by changing its coding potential, or regulate gene expression through altering alternative splicing, microRNA processing, or RNA interference (*Deffit and Hundley, 2016*; *Nishikura, 2016*). Human and mouse each has three ADARs: ADAR1, ADAR2 and ADAR3 (*Chen et al., 2000*; *Kim et al., 1994*; *Melcher et al., 1996*). ADAR1 and ADAR2 possess deaminase activity and catalyze the A-to-I

\*For correspondence:
bochen@uchc.edu

**Competing interests:** The authors declare that no competing interests exist.

conversion (*Tan et al., 2017*), whereas ADAR3 is catalytically inactive with regulatory roles in RNA editing (*Nishikura, 2016*). Millions of A-to-I editing sites have been detected in the human transcriptome through RNA-seq, with the vast majority of them found in non-coding regions (*Nishikura, 2016*). Biological effects of RNA editing at coding regions have been revealed for a variety of genes, including those encoding ligand- and voltage-gated ion channels and G protein-coupled receptors (*Bhalla et al., 2004*; *Brusa et al., 1995*; *Burns et al., 1997*; *Gonzalez et al., 2011*; *Huang et al., 2012*; *Lomeli et al., 1994*; *Palladino et al., 2000*; *Rula et al., 2008*; *Sommer et al., 1991*; *Streit et al., 2011*). However, little is known about the roles of RNA editing in non-coding regions (*Nishikura, 2016*).

In a genetic screen for suppressors of a sluggish phenotype caused by expressing a hyperactive SLO-2 in worms, we isolated mutants of several genes, including *adr-1*, which encodes one of two ADARs in *C. elegans* (ADR-1 and ADR-2). While ADR-2 has deaminase activity and plays an indispensable role in the A-to-I conversion, ADR-1 is catalytically inactive but can promote RNA editing by binding to selected target mRNA and tethering ADR-2 to RNA substrates (*Ganem et al., 2019*; *Rajendren et al., 2018*; *Washburn et al., 2014*). We found that loss-of-function (*lf*) mutations of *adr-1* inhibit SLO-2 function through impairing RNA editing of *scyl-1*, which encodes an orthologue of human and mouse SCYL1. In *adr-1(lf)* mutants, a lack of A-to-I conversion at a specific site in *scyl-1* 3'-UTR causes reduced *scyl-1* expression. Knockout of *scyl-1* severely reduces SLO-2 current in worms whereas coexpression of SCYL1 with human Slack in *Xenopus* oocytes greatly augments channel activity. These results suggest that SCYL-1/SCYL1 proteins likely play an evolutionarily conserved role in physiological functions of Slo2 channels. Mutations or knockout mammalian SCYL1 may cause neural degeneration, intellectual disabilities, and liver failure, but the underlying mechanisms are unclear (*Lenz et al., 2018*; *Li et al., 2019*; *Shohet et al., 2019*; *Spagnoli et al., 2019*). The revelation of SCYL-1/SCYL1 as a protein important to Slo2 channels suggests a potential link between diseases caused by SCLY1 mutations and Slo2 channel functions.

## Results

### *adr-1* mutants suppress sluggish phenotype of *slo-2(gf)*

In a genetic screen for mutants that suppressed a sluggish phenotype caused by an engineered hyperactive or gain-of-function (*gf*) SLO-2 (*Liu et al., 2018*), we isolated two mutants (*zw80* and *zw81*) of the *adr-1* gene, as revealed by analyses of whole-genome sequencing data. *zw80* and *zw81* carry nonsense mutations leading to premature stops at tryptophan (W) 366 and W33, respectively (*Figure 1A*). *slo-2(gf)* worms showed greatly decreased locomotion speed compared with wild type, and this phenotype was substantially alleviated in *slo-2(gf);adr-1(lf)* double mutants (*Figure 1B*). To confirm that the suppression of *slo-2(gf)* phenotype resulted from mutations of *adr-1* rather than that of another gene, we created a new *adr-1* mutant allele (*zw96*) by introducing a premature stop codon at serine (S) 333 (*Figure 1A*) using the CRISPR/Cas9 approach. The sluggish phenotype of *slo-2(gf)* was similarly suppressed by *adr-1(zw96)*, which, by itself, did not enhance locomotion speed (*Figure 1B*). Expression of wild-type *adr-1* under the control of the pan-neuronal *rab-3* promotor (P*rab-3*) in *slo-2(gf);adr-1(zw96)* unmasked the sluggish phenotype (*Figure 1B*). These results indicate that the sluggish phenotype of *slo-2(gf)* is mainly caused by SLO-2 hyperactivity in neurons, and that neuronal function of SLO-2(*gf*) depends on ADR-1.

In *C. elegans*, cholinergic motor neurons control body-wall muscle cells by producing bursts of postsynaptic currents (PSC bursts) (*Liu et al., 2014*). To determine how *adr-1* mutants might alleviate the *slo-2(gf)* locomotion defect, we recorded voltage-activated whole-cell currents from a representative cholinergic motor neuron (VA5) and postsynaptic currents from body-wall muscle cells in wild type, *slo-2(gf)*, *slo-2(gf);adr-1(zw96)*, and *slo-2(gf);adr-1(zw96)* with *adr-1* rescued in neurons. Compared with wild type, the *slo-2(gf)* strain displayed much larger outward currents, and greatly decreased PSC burst frequency, duration and charge transfer (*Figure 1C and D*). These phenotypes of *slo-2(gf)* were mostly suppressed in the *slo-2(gf);adr-1(zw96)* strain (*Figure 1C and D*), suggesting that *adr-1(lf)* alleviated the sluggish phenotype through inhibiting SLO-2(*gf*). In addition, expression of wild-type *adr-1* in neurons of *slo-2(gf);adr-1(zw96)* unmasked the effects of *slo-2(gf)* on VA5 whole-cell currents and PSC bursts (*Figure 1C and D*). These observations suggest that the

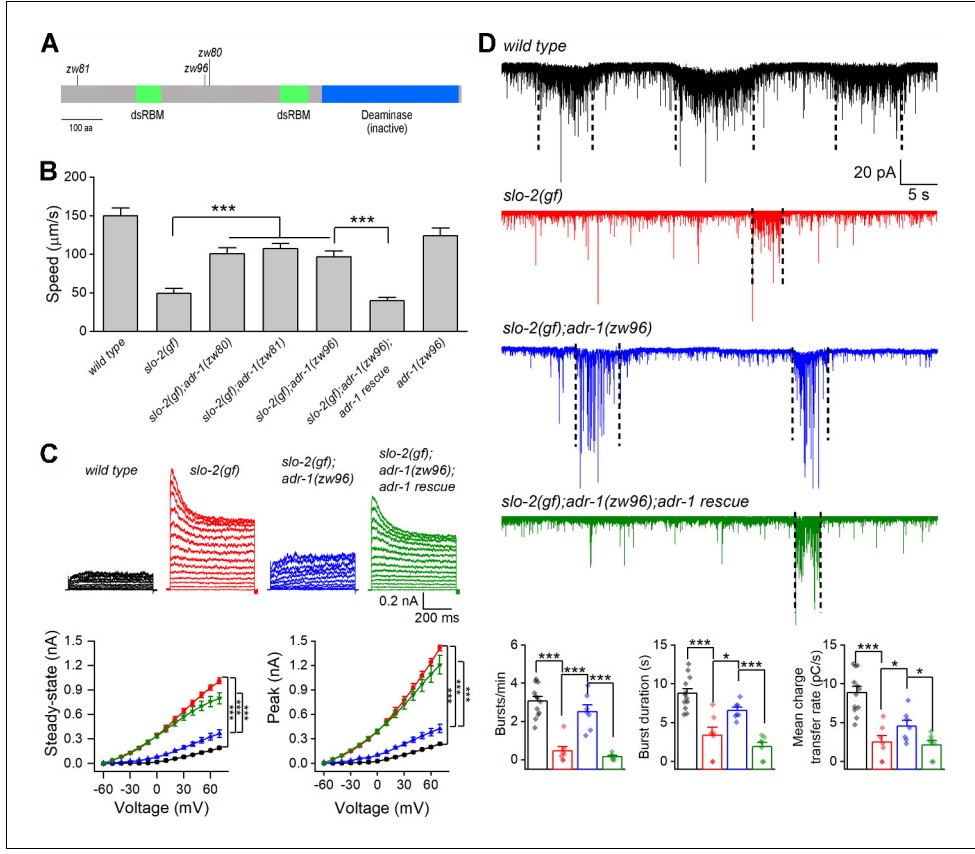

**Figure 1.** Loss-of-function mutations of *adr-1* suppress phenotypes caused by a hyperactive SLO-2. (A) Diagram of ADR-1 domain structures and locations of the non-sense mutations in the *adr-1* mutants. ADR-1 has two double-stranded RNA-binding motifs (dsRBM) and a pseudodeaminase domain. (B) Mutations of *adr-1* mitigated an inhibitory effect of hyperactive or gain-of-function (*gf*) SLO-2 on locomotion through acting in neurons. *adr-1* rescue was achieved by expressing GFP-tagged wild-type ADR-1 in neurons under the control of P*rab-3* (same in C and D). Sample sizes were 10–12 in each group. (C) *adr-1(zw96)* reduced an augmenting effect of *slo-2(gf)* on motor neuron whole-cell outward currents. Pipette solution I and bath solution I were used. Sample sizes were 7 *wild type*, 8 *slo-2(gf)*, 9 *slo-2(gf);adr-1(zw96)*, and 8 *slo-2(gf);adr-1(zw96)* rescue. (D) *adr-1(zw96)* mitigated an inhibitory effect of *slo-2(gf)* on postsynaptic current (PSC) bursts at the neuromuscular junction. The vertical dotted lines over the sample traces mark PSC bursts, which are defined as an apparent increase in PSC frequency accompanied by a sustained current (downward baseline shift) lasting >3 s. Pipette solution II and bath solution I were used. Sample sizes were 12 *wild type*, and 7 in each of the remaining groups. All values are shown as mean ± SE. The asterisks indicate statistically significant differences between indicated groups (*p<0.05, ***p<0.001) based on either two-way (C) or one-way (D) ANOVA with Tukey's post hoc tests.

The online version of this article includes the following source data for figure 1:

**Source data 1.** Raw data and numerical values for data plotted in *Figure 1*.

suppressing effect of *adr-1(lf)* on the *slo-2(gf)* sluggish phenotype was likely due to reduced SLO-2 activities in motor neurons.

We suspected that the suppression of SLO-2(*gf*) by *adr-1(lf)* resulted from deficient RNA-editing. If so, *adr-2(lf)* might also suppress the sluggish phenotype of *slo-2(gf)* because ADR-2 is required for RNA editing. Indeed, the sluggish phenotype of *slo-2(gf)* worms was substantially alleviated in *slo-2(gf);adr-2(lf)* double mutants (*Figure 2A*), and the augmenting effect of *slo-2(gf)* on VA5 whole-cell outward currents was mostly eliminated by *adr-2(lf)* (*Figure 2B*). Furthermore, *adr-2(lf)* brought VA5 whole-cell currents below the wild-type level (*Figure 2B*), which presumably resulted from reduced activities of wild-type SLO-2. These results suggest that RNA editing is important to SLO-2 function in neurons.

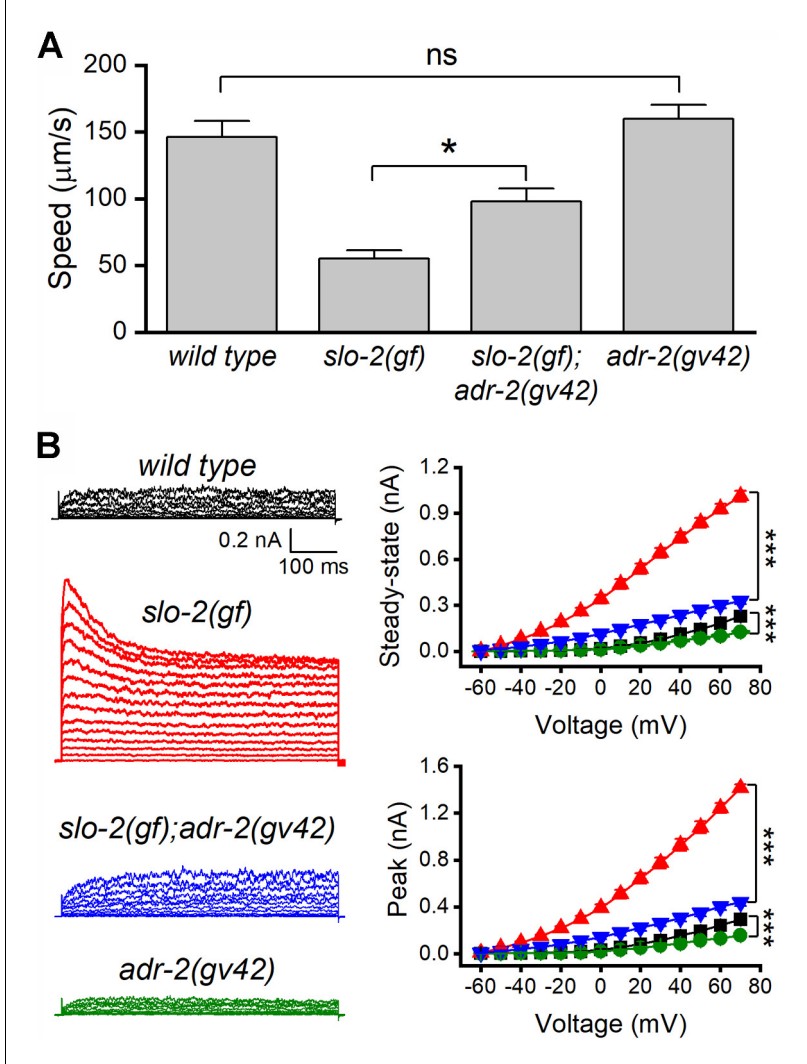

**Figure 2.** Loss-of-function mutation of *adr-2* suppressed the effects of gain-of-function (*gf*) *slo-2* on locomotion and motor neuron whole-cell currents. (A) *adr-2(gv42)* alleviated an inhibitory effect of *slo-2(gf)* on locomotion speed. The sample size was 10–12 in each group. (B) *adr-2(gv42)* largely reversed an augmenting effect of *slo-2(gf)* on whole-cell currents in VA5 motor neuron. Sample sizes were 11 *wild type*, 8 *slo-2(gf)*, 11 *slo-2(gf);adr-2(gv42)*, and 10 *adr-2(gv42)*. All data are shown as mean ± SE. Pipette solution I and bath solution I were used. The asterisks indicate statistically significant differences (*p<0.05; ***p<0.001) whereas 'ns' stands for 'no significant difference' between the indicated groups based on either one-way (A) or two-way (B) ANOVA with Tukey's post hoc tests.

The online version of this article includes the following source data for figure 2:

**Source data 1.** Raw data and numerical values for data plotted in *Figure 2*.

## ADR-1 is expressed in neurons and localized in the nucleus

The expression pattern of *adr-1* was examined by expressing GFP under the control of *adr-1* promoter (P*adr-1*). In transgenic worms, strong GFP expression was observed in the nervous system, including ventral cord motor neurons and many neurons in the head and tail, while weak GFP expression was observed in the intestine and body-wall muscles (*Figure 3A*). We then examined the subcellular localization pattern of ADR-1 by expressing GFP-tagged full-length ADR-1 (ADR-1::GFP) under the control of P*rab-3*. We found that ADR-1::GFP is localized in the nucleus, as indicated by its colocalization with the mStrawberry-tagged nucleus marker HIS-58 (*Liu et al., 2018*) in ventral cord motor neurons (*Figure 3B*).

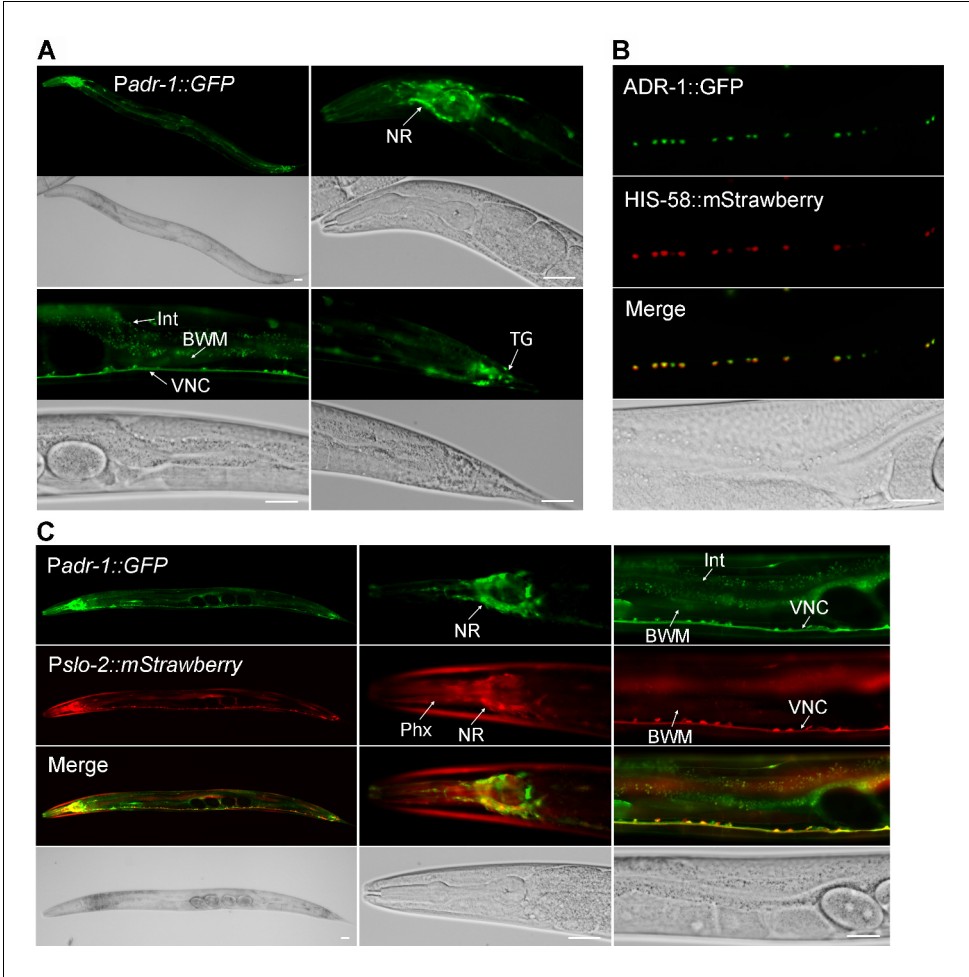

**Figure 3.** ADR-1 is coexpressed with SLO-2 in many neurons and localized in the nucleus. (**A**) Expression of an *adr-1* promoter (P*adr-1*)::GFP transcriptional fusion in worms resulted in strong GFP signal in many neurons (NR, nerve ring; VNC, ventral nerve cord; TG, tail ganglion) and weak GFP signal in body-wall muscles (BWM) and intestine (Int). (**B**) GFP-tagged ADR-1 (ADR-1::GFP) colocalized with a mStrawberry-tagged HIS-58 nucleus marker, as indicated by fluorescence images of VNC motor neurons. (**C**) *adr-1* and *slo-2* are co-expressed in many neurons but show differential expressions in the pharynx (Phx) and Int. Scale bar = 20 μm in in all panels.

To determine whether *adr-1* is co-expressed with *slo-2*, we crossed the P*adr-1::GFP* transgene into an existing strain expressing P*slo-2*::mStrawberry (*Liu et al., 2018*). We found that the expression patterns of *adr-1* and *slo-2* overlapped extensively in the nervous system (*Figure 3C*). For example, the majority of ventral cord motor neurons and numerous head neurons were colabeled by GFP and mStrawberry (*Figure 3C*). The occasional non-overlapping expressions of GFP and mStrawberry in ventral cord motor neurons probably resulted from mosaic expression of the transgenes.

## ADR-1 regulates neurotransmitter release through SLO-2

SLO-2 is the primary conductor of delayed outward currents in *C. elegans* cholinergic motor neurons (*Liu et al., 2014*). We wondered whether the function of native SLO-2 channels in motor neurons depends on ADR-1. Consistent with our previous report (*Liu et al., 2014*), VA5 delayed outward currents were dramatically smaller and VA5 resting membrane potential was much less hyperpolarized in *slo-2(lf)* than wild type. While *adr-1(lf)* also caused significantly decreased outward currents and less hyperpolarized resting membrane potential in VA5, it did not produce additive effects when combined with *slo-2(lf)* (*Figure 4A–C*). These results suggest that *adr-1(lf)* affects motor neuron outward currents and resting membrane potential through SLO-2.

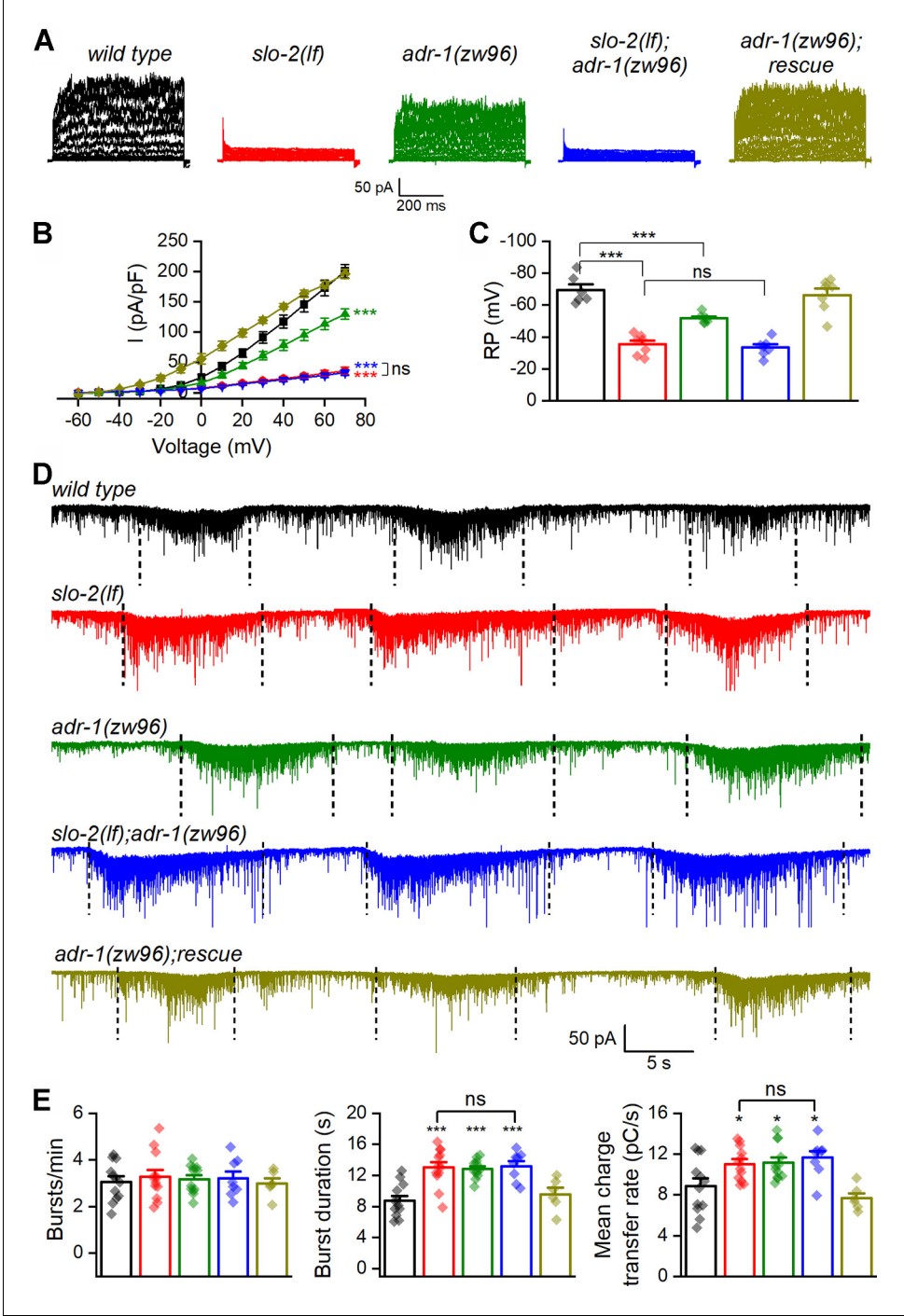

**Figure 4.** ADR-1 contributes to motor neuron whole-cell currents and regulates postsynaptic current (PSC) bursts through SLO-2. (**A**) Representative VA5 whole-cell current traces. (**B**) Current (I) - voltage relationships of the whole-cell currents. Sample sizes were 8 *wild type*, 7 *slo-2(lf)*, 9 *adr-1(zw96)*, 7 *slo-2(lf);adr-1(zw96)*, and 9 *adr-1 (zw96)* rescue. (**C**) Resting membrane potentials of VA5. Sample sizes were 6 *wild type*, and 7 in each of the remaining groups. (**D**) Representative traces of spontaneous PSCs with PSC bursts marked by vertical dotted lines. (**E**) Comparisons of PSC burst properties. Sample sizes were 8 *slo-2(lf);adr-1(zw96)*, 6 *adr-1(zw96)* rescue, and 12 in each of the remaining groups. All values are shown as mean ± SE. The asterisks indicate statistically significant differences (*p<0.05, ***p<0.001) compared with *wild type* whereas 'ns' stands for no significant difference between the indicated groups based on either two-way (**B**) or one-way (**C and E**) ANOVA with Tukey's post hoc tests. Pipette solution I and bath solution I were used in (**A**) and (**C**). Pipette solution II and bath solution I were used in (**D**).

*Figure 4 continued on next page*

*Figure 4 continued*

The online version of this article includes the following source data and figure supplement(s) for figure 4:

**Source data 1.** Raw data and numerical values for data plotted in *Figure 4*.

**Figure supplement 1.** Comparison of *slo-2* transcript level between *wild type* and *adr-1* mutant.

We next determined whether *adr-1(lf)* also alters PSC bursts. We found that *adr-1(lf)* caused an increase in the duration and mean charge transfer rate of PSC bursts without altering the burst frequency compared with wild type (*Figure 4D and E*). These phenotypes of *adr-1(lf)* were similar to those of *slo-2(lf)* and did not become more severe in the double mutants (*Figure 4D and E*), suggesting that ADR-1 modulates neurotransmitter release through SLO-2. The similar effects of *adr-1 (lf)* and *slo-2(lf)* on PSC bursts are in contrast to their differential effects on VA5 outward currents and resting membrane potential. This discrepancy suggests that reducing SLO-2 activity beyond a certain threshold may produce similar effects on PSC bursts as does *slo-2(lf)*.

## ADR-1 regulates SLO-2 function through SCYL-1

Given that our results suggest that RNA editing is important to SLO-2 function, we determined whether *adr-1(lf)* causes deficient editing or decreased expression of *slo-2* mRNA by comparing RNA-seq data between *adr-1(lf)* and wild type. The *adr-1(zw96)* allele was chosen for these analyses to avoid complications by potential mutations of other genes introduced into the genome during the generation of the other *adr-1* mutants (*zw80* and *zw81*). Unexpectedly, no RNA editing event was detected in *slo-2* transcripts, and *slo-2* mRNA level was similar between wild type and the *adr-1* mutant (*Figure 4—figure supplement 1*). These results suggest that ADR-1 might regulate SLO-2 function through RNA editing of another gene.

A previous study identified 270 high-confidence editing sites in transcripts of 51 genes expressed in *C. elegans* neurons (*Washburn et al., 2014*). We suspected that the putative molecule mediating the effect of ADR-1 on SLO-2 is encoded by one of these genes, and the mRNA level of this gene is reduced in *adr-1(lf)*. Therefore, we compared transcript expression levels of these genes (excluding those encoding transposons) in our RNA-Seq data between wild type and *adr-1(zw96)*. Most of these genes showed either no decrease or only a small decrease in expression, but two of these genes, *rncs-1* and *scyl-1*, were reduced greatly in *adr-1(lf)* compared with wild type (*Figure 5*). *rncs-1* is not a conceivable candidate for the putative SLO-2 regulator because it is a non-coding gene expressed in the hypodermis and vulva (*Hellwig and Bass, 2008*). We therefore focused our analyses on *scyl-1*, which encodes an orthologue of mammalian SCYL1 important to neuronal function and survival (*Pelletier, 2016*). Like its mammalian homologs, SCYL-1 has an amino-terminal kinase domain that lacks residues critical to kinase activity, and a central domain containing five HEAT repeats (HEAT for *Hun*-tingtin, *e*longation factor 3, protein phosphatase 2*A*, yeast kinase *T*OR1) (*Pelletier, 2016*). SCYL-1 shares 38% identity and 60% similarity with human SCYL1. Notably, amino acid sequence in the HEAT domain, which is often highly degenerative (*Pelletier, 2016*), shows a very high level of sequence homology (53% identity and 76% similarity) between these two proteins (*Figure 5—figure supplement 1*).

We examined the expression pattern of *scyl-1* by expressing GFP under the control of *scyl-1* promoter (P*scyl-1*). Because another gene (*lap-2*) resides ~2 kb upstream of *scyl-1* (www.wormbase.org), we first expressed GFP under the control of 2 kb P*scyl-1*. However, no GFP signal was detected in transgenic worms (not shown). We then used an in vivo homologous recombination approach to express a P*scyl-1::GFP* transcriptional fusion that included a much longer sequence upstream of the *scyl-1* initiation site. Specifically, a 0.5 kb genomic fragment upstream of the *scyl-1* initiation site was fused to GFP in a plasmid, which was co-injected with a fosmid covering part of the *scyl-1* coding region and 32 kb sequence upstream of the initiation site into wild type worms. In vivo homologous recombination between the plasmid and the fosmid is expected to result in a *promoter::GFP* transcriptional fusion that includes all the upstream genomic sequence in the fosmid. Transgenic worms from coinjection of the plasmid and the fosmid showed GFP signal in a variety of cells (*Figure 6*), suggesting distant upstream sequences are required for *scyl-1* expression. To determine how the expression pattern of *scyl-1* correlates with that of *slo-2*, we crossed the transgene into the P*slo-2*::mStrawberry strain, and examined the expression patterns of GFP and mStrawberry. Co-expression

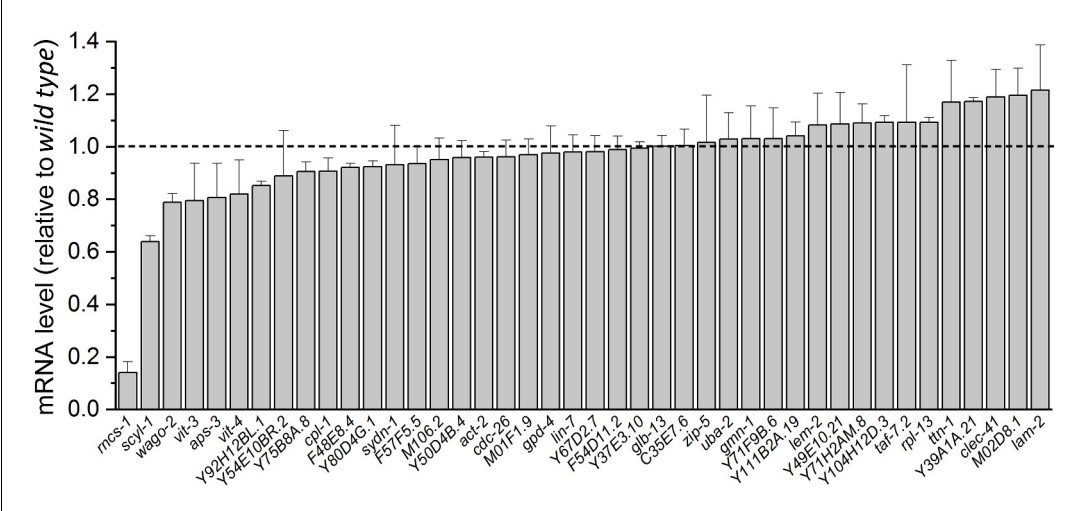

**Figure 5.** Normalized transcript expression levels of selected genes in *adr-1(zw96)* mutant. The genes were selected based on the detection of ADR-1-dependent RNA editing events in their transcripts reported in an earlier study (*Washburn et al., 2014*). Transcript expression level of each gene in the mutant is normalized by that in the wild type. Shown are mean ± SE from three biological replicates of RNA-seq experiments.

The online version of this article includes the following source data and figure supplement(s) for figure 5:

**Source data 1.** Raw data and numerical values for data plotted in *Figure 5*.

**Figure supplement 1.** Alignment of amino acid sequences between *C. elegans* SCYL-1 (W07G4.3, www.wormbase.org) and human SCYL1 (hSCYL1, GenBank: NP_065731.3).

of *scyl-1* and *slo-2* was observed in many ventral cord motor neurons (*Figure 6*). However, most other neurons expressing *slo-2* (e. g. head and tail neurons) did not appear to express *scyl-1*. In addition, *scyl-1* expression was detected in some cells that did not express *slo-2*, including the excretory cell, spermatheca, uterine ventral cells, and intestinal cells (*Figure 6*).

We next determined whether SCYL-1 is important to SLO-2 function. To this end, we created a mutant, *scyl-1(zw99)*, by introducing a stop codon after isoleucine 152 using the CRISPR/Cas9 approach, and examined the effect of this mutation on VA5 delayed outward currents. *scyl-1(zw99)* showed a substantial decrease in VA5 outward currents compared with wild type; this phenotype could be rescued by expressing wild-type SCYL-1 in neurons, and was non-additive with that of either *slo-2(lf)* or *adr-1(zw96)* (*Figure 7A*). These results suggest that SCYL-1 and ADR-1 likely act in a common pathway to contribute to SLO-2 function.

The decrease of delayed outward currents in *scyl-1(lf)* could have resulted from either reduced expression or reduced function of SLO-2. We first determined whether *scyl-1(lf)* alters SLO-2 expression by crossing a stable (near 100% penetrance) P*rab-3*::SLO-2::GFP transgene from an existing transgenic strain of wild-type genetic background (*Liu et al., 2018*) into *scyl-1(zw99)*, and comparing GFP signal between the two strains. We found that GFP signal in the ventral nerve cord was similar between wild type and the *scyl-1* mutant (*Figure 7B*), suggesting that SCYL-1 does not regulate SLO-2 expression. We then determined whether SCYL-1 regulates SLO-2 function by obtaining inside-out patches from VA5 and analyzing SLO-2 single-channel properties. SLO-2 showed >50% decrease in open probability ($P_o$) without a change of single-channel conductance in *scyl-1(zw99)* compared with wild type, and this mutant phenotype was completely rescued by neuronal expression of wild-type SCYL-1 (*Figure 8A*). Analyses of single-channel open and closed events revealed that SLO-2 has at least two open states and three closed states, and that the decreased $P_o$ of SLO-2 in *scyl-1(lf)* mainly resulted from shorter and fewer long openings (*Figure 8B*) and a larger proportion of the longest closings (*Figure 8C*).

The observed effects of *scyl-1(lf)* on SLO-2 single-channel properties suggest that SCYL-1 may physically interacts with SLO-2. To address this possibility, we performed bimolecular fluorescence complementation (BiFC) (*Hu et al., 2002*) and coimmunoprecipitation assays, which indicate whether these two proteins are physically very close in vivo and whether they coexist in a molecular complex,

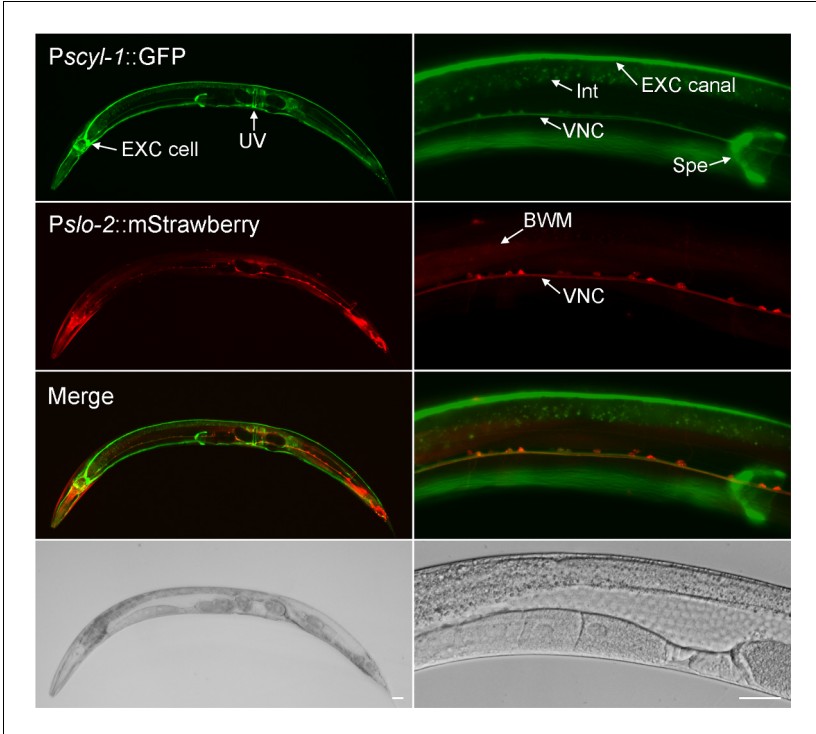

**Figure 6.** *scyl-1* and *slo-2* are coexpressed in ventral cord motor neurons but differentially expressed in other cells. In transgenic worms coexpressing P*scyl-1::GFP* and P*slo-2::mStrawberry* transcriptional fusions, GFP signal was observed in ventral nerve cord (VNC) motor neurons, the large H-shaped excretory (EXC) cell, uterine ventral (UV) cells, and spermatheca (Spe) while mStrawberry signal was detected in VNC motor neurons, body-wall muscles (BMW), and many other neurons. Scale bar = 20 μm.

respectively. In both assays, we determined whether full-length SCYL-1 interacts with either the full-length, the amino-terminal portion (amino acids 1–317), or the carboxyl-terminal portion (amino acids 318–1107) of SLO-2 (*Figure 9A*). In the BiFC assay, SCYL-1 and SLO-2 were fused to the N- and C-terminal portions of YFP (YFPa and YFPc), respectively (*Figure 9A*). A detection of YFP signal would indicate physical closeness of the two fusion proteins. We observed YFP fluorescence in ventral cord motor neurons when either the full-length or the C-terminal of SLO-2 was used but not when the N-terminal was used in the assays (*Figure 9B*). The coimmunoprecipitation assay was performed with homogenates of worms expressing HA-tagged SCYL-1 and GFP-tagged SLO-2. We found that the SCYL-1 immunoprecipitated with either full-length SLO-2 or SLO-2 C-terminal but not SLO-2 N-terminal (*Figure 9C*), which is in agreement with the BiFC results. Thus, both the BiFC and coimmunoprecipitation results suggest that SCYL-1 physically interacts with SLO-2, and this interaction is mediated by SLO-2 carboxyl terminal.

## *scyl-1* expression depends on RNA editing at a specific 3'-UTR site

Our RNA-Seq data revealed eight high-frequency (>15%) adenosine-to-guanosine editing sites in *scyl-1* transcripts of wild type (*Figure 10A*). All these editing sites are located within a predicted 746 bp hair-pin structure in the 3' end of *scyl-1* pre-mRNA, which contains an inverted repeat with >98% complementary base pairing (*Figure 10B*). Interestingly, RNA editing at only one of these sites was significantly deficient (by 74%) in *adr-1(zw96)* compared with wild type (*Figure 10A*). Sanger sequencing of *scyl-1* mRNA and the corresponding genomic DNA from wild type, *adr-1(zw96)*, and *adr-2(gv42)* showed that RNA editing at this site was deficient in both the *adr-1* and *adr-2* mutants whereas that at an adjacent site was deficient only in the *adr-2* mutant (*Figure 10C*), suggesting that RNA editing at the site impaired by *adr-1(lf)* might be important to *scyl-1* expression. To address this possibility, we created transgenic worms expressing a P*rab-3::GFP* transcriptional fusion (*wp1923*), in which a genomic DNA fragment covering part of the last exon of *scyl-1* and 5 kb

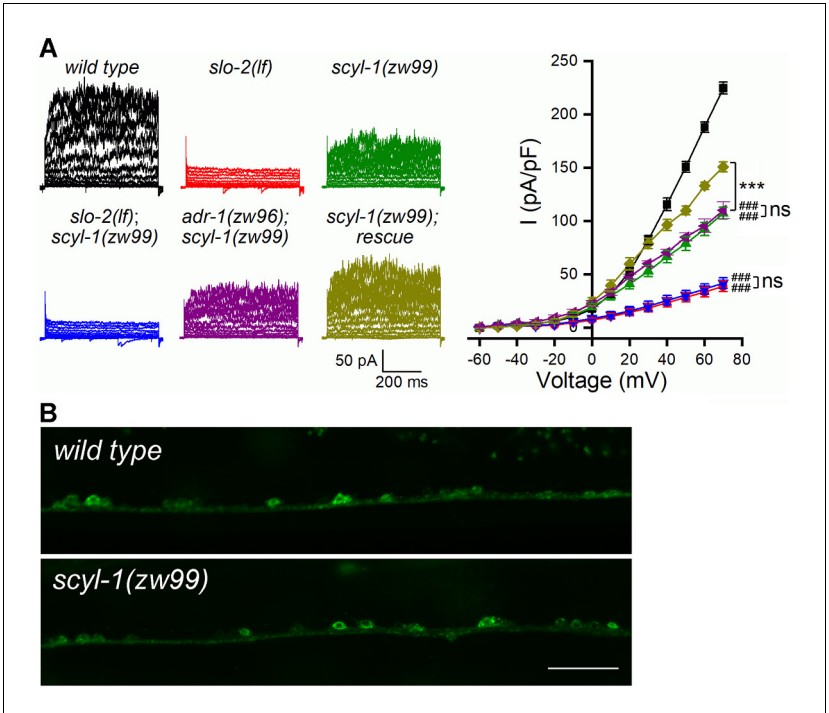

**Figure 7.** SCYL-1 contributes to motor neuron outward currents through SLO-2. (**A**) Sample whole-cell current traces of VA5 motor neurons and the current-voltage relationships. Sample sizes were 9 *wild type*, 7 *slo-2(lf)*, 10 *scyl-1(zw99)*, 7 *slo-2(lf);scyl-1(zw99)*, 7 *adr-1(zw96);scyl-1(zw99)*, and 7 *scyl-1(zw99)* rescue. The rescue strain was created by expressing wild-type *scyl-1* under the control of P*rab-3*. All values are shown as mean ± SE. The asterisks (***) and pound signs (###) indicate statistically significant differences (p<0.001) between the indicated groups and from wild type, respectively, whereas 'ns' stands for no significant difference between the indicated groups (two-way ANOVA with Tukey's post hoc tests). (**B**) GFP signal in ventral cord motor neurons was indistinguishable between *wild type* and *scyl-1(zw99)* worms expressing GFP-tagged full-length SLO-2 under the control of P*rab-3*. Scale bar = 20 μm.

The online version of this article includes the following source data for figure 7:

**Source data 1.** Raw data and numerical values for data plotted in *Figure 7*.

downstream sequence was fused in-frame to the 3'-end of GFP coding sequence (*Figure 10D*). We also created transgenic worms expressing a modified plasmid (*wp1924*), in which adenosine (A) at the specific ADR-1-dependent editing site was changed to guanosine (G) to mimic the edited nucleotide (*Figure 10D*). GFP signal was observed in worms harboring *wp1924* but no GFP signal was detected in worms harboring *wp1923* (*Figure 10E*). While observation of GFP signal in the *wp1924* strain was expected, the complete absence of GFP signal in the *wp1923* strain caught us by surprise. To better understand the role of ADR-1 in *scyl-1* mRNA expression, we integrated the *wp1924* transgene into the wild-type genome, crossed it into *adr-1(zw96)*, and compared GFP signal in ventral cord motor neurons between wild type and the mutant. GFP signal was ~50% weaker in the mutant than wild type (*Figure 10F and G*). Because the transgene mimicked the edited 3'-UTR sequence of *scyl-1* mRNA, the weaker GFP signal in the mutant than wild type suggests that ADR-1 can also increase *scyl-1* mRNA level through a post-editing effect.

If ADR-1 does enhance *scyl-1* mRNA level through a post-editing mechanism in addition to RNA editing, it likely perform this function through interacting with some other proteins, and such proteins might be identified by screening for mutants showing decreased GFP signal from the *wp1924* transgene. Indeed, eleven mutants with reduced GFP expression were isolated from ~5000 mutagenized haploid genomes in a pilot genetic screen, as shown by two examples (*Figure 10—figure supplement 1*). The deficiency of GFP expression in these mutants was related to the function of *scyl-1* 3'-UTR because GFP expression from a control transgene with *unc-10* 3'-UTR was not

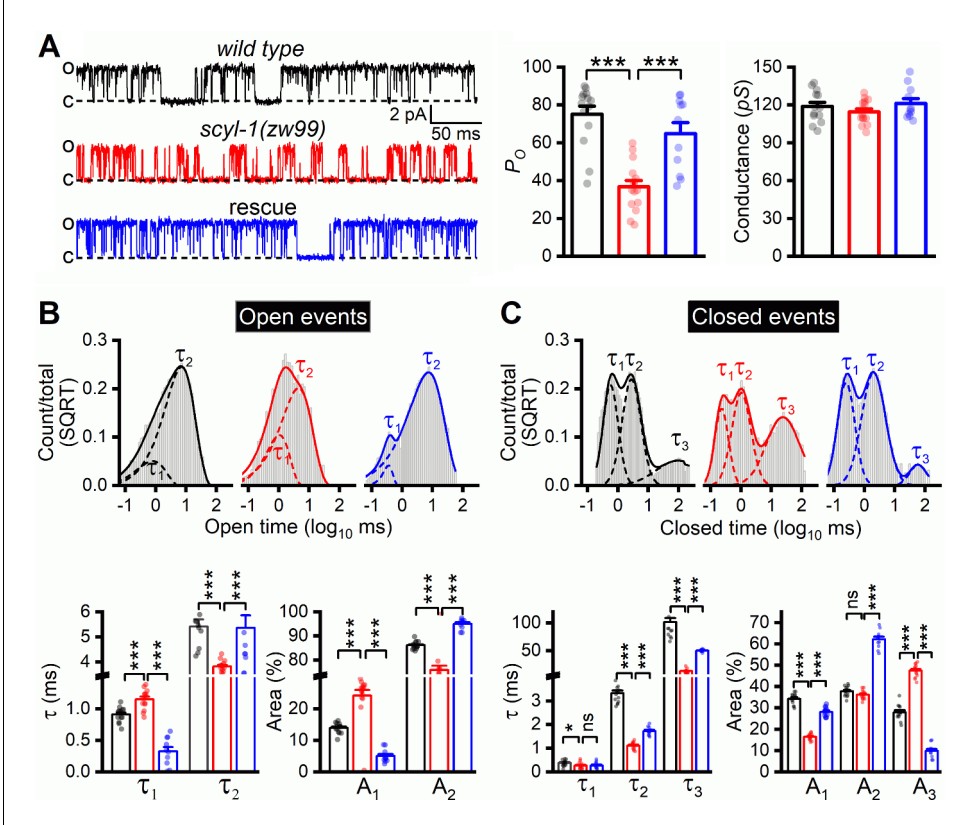

**Figure 8.** Single-channel open probability ($P_o$) of SLO-2 is decreased in *scyl-1* mutant. (**A**) Representative SLO-2 single-channel currents from inside-out patches of the VA5 motor neuron, and comparisons of $P_o$ and single-channel amplitude between *wild type* (n = 14), *scyl-1(zw99)* (n = 15), and *scyl-1(zw99)* rescued by expressing wild-type *scyl-1* in neurons under the control of P*rab-3* (n = 11). (**B and C**) Fitting of open and closed dwell time histogram to exponentials, and comparisons of τ values and relative areas (**A**) of the fitted components. All the open and closed events of each group were pooled together to plot the dwell time histograms. Statistical comparisons shown below were based on the mean τ values of individual recordings with each dot representing the mean value of one recording. Pipette solution III and bath solution II were used. All values are shown as mean ± SE. The asterisks indicate a significant difference between the indicated groups (*p<0.05, ***p<0.001, one-way ANOVA with Tukey's post hoc tests).

The online version of this article includes the following source data for figure 8:

**Source data 1.** Raw data and numerical values for data plotted in *Figure 8*.

compromised in these mutants (*Figure 10—figure supplement 1*). These results favor the notion that ADR-1 may also interact with other molecules after RNA editing to promote *scyl-1* expression.

## Human Slo2.2/Slack is regulated by SCYL1

The HEAT domain of SCYL proteins is important to protein-protein interactions but generally varies considerably in amino acid sequence for interactions with specific protein partners (*Yoshimura and Hirano, 2016*). The high level of sequence homology of the HEAT domain between mammalian SCYL1 and worm SCYL-1 (*Figure 5—figure supplement 1*) promoted us to test whether mammalian Slo2.2/Slack is also regulated by SCYL1. We expressed human Slo2.2 (hSlo2.2) either alone or together with mouse SCYL1 in *Xenopus* oocytes, and analyzed hSlo2.2 single-channel properties in inside-out patches. SCYL1 increased hSlo2.2 $P_o$ greatly without altering the single-channel conductance (*Figure 11A*). The channel has at least two open states and three closed states (*Figure 11B–D*). While events of both the open states and the two shorter closed states were numerous, those of the longest closed state were infrequent. Nevertheless, the longest closed state had a major impact on $P_o$ because of its rather long duration. Dwell time analyses indicate that SCYL1 increased hSlo2.2

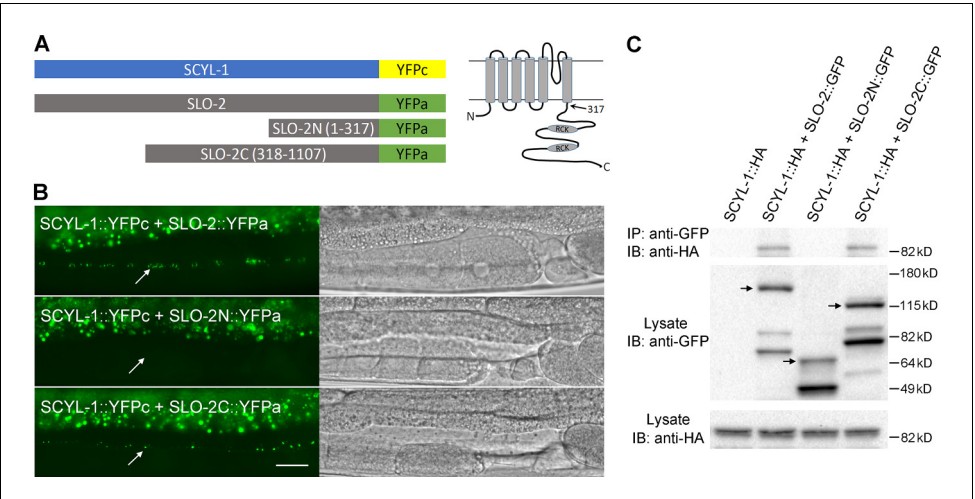

**Figure 9.** SCYL-1 physically interacts with SLO-2 in neurons. (**A**) Diagrams of the various fusion proteins used in the BiFC assays (*left*) and of SLO-2 membrane topology (*right*). The arrow indicates the split site for SLO-2N and SLO-2C fusions. RCK, regulator of conductance for K$^+$. (**B**) YFP signal was detected when SCYL-1 was coexpressed with either full-length or the carboxyl terminal portion of SLO-2 but not with the amino terminal portion of SLO-2. Shown are representative fluorescent images of the ventral nerve cord (indicated by arrows) with corresponding DIC images. The bright signals at the top of each fluorescence image was from auto-fluorescence of the intestine. Scale bar = 20 μm. (**C**) SCYL-1 co-immunoprecipitates with full-length SLO-2 and SLO-2C but not SLO-2N. IP, immunoprecipitation; IB, immunoblot. The molecular masses of the protein standard are indicated on the right. Note that multiple bands are seen in the lanes loaded with GFP fusions. The bands that match predicted molecular masses of SLO-2::GFP, SLO-2N::GFP, and SLO-2C::GFP fusions are indicated with arrows, respectively. The other bands likely resulted from cleaved or partially translated GFP fusion proteins.

$P_o$ mainly by increasing the duration and proportion of the longer open state, and shortening the duration of the longest closed state (*Figure 11B–D*). The overall effect of SCYL1 on hSlo2.2 $P_o$ is similar to that of SCYL-1 on SLO-2 $P_o$ in wild-type worms (*Figure 8*). Taken together, these results suggest that a physiological function of both mammalian SCYL1 and worm SCYL-1 is to enhance Slo2 channel activities.

## Discussion

This study shows that both ADR-1 and SCYL-1 are critical to SLO-2 physiological function in neurons. While ADR-1 enhances SLO-2 function indirectly through promoting SCYL-1 expression, SCYL-1 regulates SLO-2 through physical interactions. These conclusions are supported by multiple lines of evidence, including the isolation of *adr-1(lf)* mutants as suppressors of SLO-2(*gf*), the inhibition of SLO-2 activities by either *adr-1(lf)* or *scyl-1(lf)*, the reduction of *scyl-1* transcript expression in *adr-1(lf)*, the correlation between *scyl-1* mRNA level and RNA editing at its 3'-UTR, the evidence of physical interactions between SCYL-1 and SLO-2, and the reduction of SLO-2 $P_o$ due to changes in the dwell times of open and closed events in *scyl-1(lf)*. Importantly, we found that the human Slo2.2/Slack is also regulated by SCYL1.

The biological significance of RNA editing at non-coding regions is only beginning to be appreciated. A recent study with *C. elegans* identified many neuron-specific A-to-I editing sites in the 3'-UTR of *clec-41*, and showed that elimination of these editing events by *adr-2* knockout compromises *clec-41* expression and impairs a chemotaxis behavior (*Deffit et al., 2017*). However, it remains to be determined how *clec-41* expression is controlled by these editing events, and whether the chemotaxis defect of *adr-2(lf)* mutant is caused by decreased *clec-41* expression. In the present study, we demonstrate that an A-to-I RNA editing event at the 3'UTR of *scyl-1* controls its expression, and that SCYL-1 contributes to neuronal whole-cell currents through a direct effect on SLO-2. The results of these two studies have provided a glimpse of the biological roles of 3'-UTR RNA editing in gene expression and neuronal function.

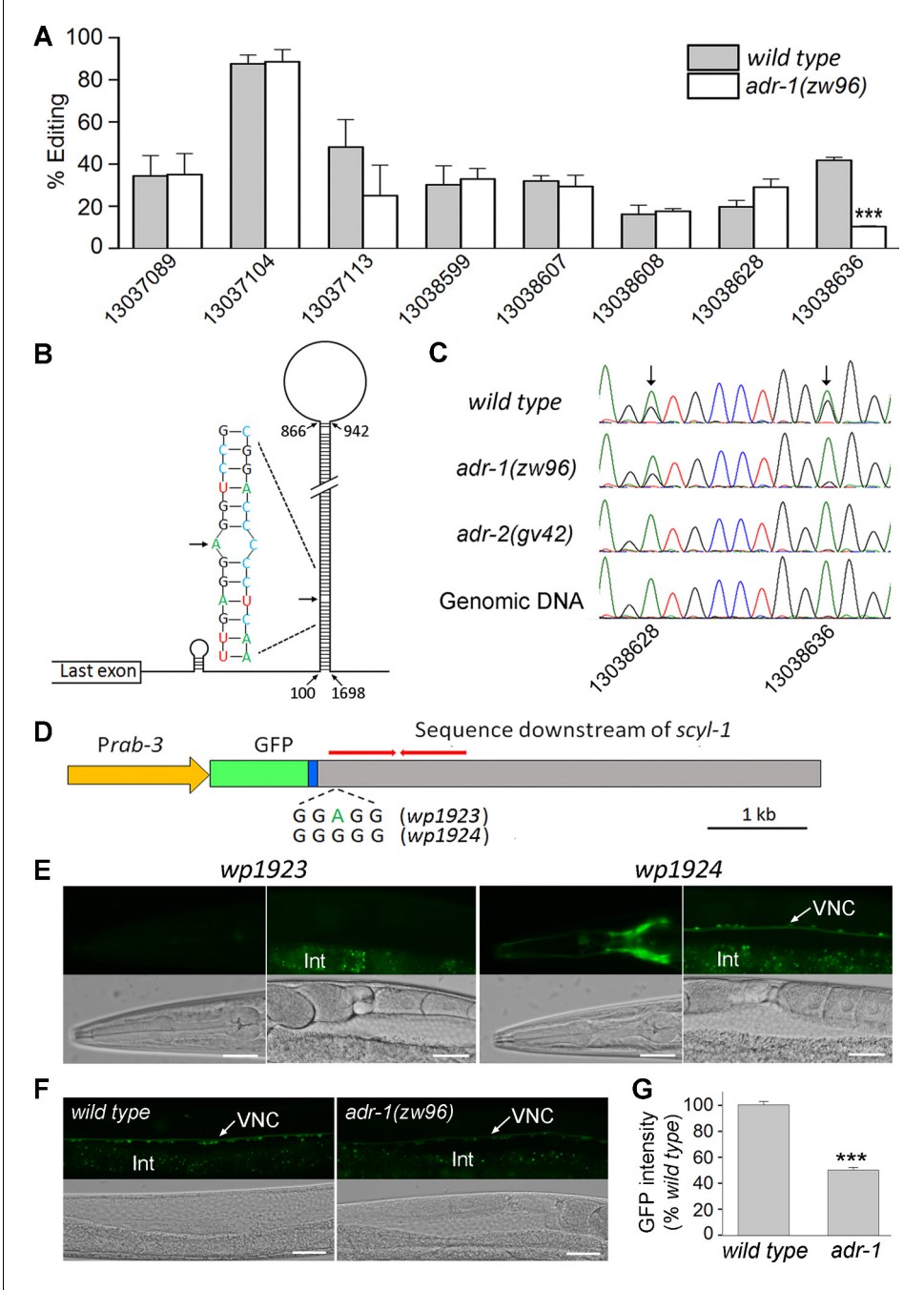

**Figure 10.** ADR-1 regulates *scyl-1* expression through RNA editing at a specific nucleotide in the 3'-UTR. (**A**) RNA editing at one out of eight highly (>15%) edited sites is severely deficient in *adr-1(zw96)* compared *wild type*. The percentage of editing was calculated by diving the number of reads containing A-I conversion by the total number of reads at each site. The x-axis indicates the positions of the edited adenosines in chromosome *V* (NC_003283). Shown are results (mean ± SE) of three RNA-seq experiments. The asterisks (***) indicate a statistically significant difference (p<0.001, unpaired *t*-test). (**B**) Diagram showing a predicted hair-pin structure in the 3' end of *scyl-1* pre-mRNA with 746 complementary base pairs. Nucleotide are numbered from the first nucleotide of the 3'-UTR. (**C**) Chromatograms of *scyl-1* mRNA 3'-UTRs of *wild type*, *adr-1(zw96)*, and *adr-2(gv42)*, and of the corresponding *wild type* genomic DNA. Two editing sites in *wild type* mRNA (indicated by arrows) display a mixture of green (adenosine) and black (guanosine) peaks. While both editing events are non-existent in *adr-2(gv42)*, only one of them is inhibited by *adr-1(zw96)*. (**D**) Diagram of two GFP reporter constructs (*wp1923* and *wp1924*) used to confirm the role of the ADR-1-dependent editing site in gene expression. GFP was placed under the control of

*Figure 10 continued on next page*

*Figure 10 continued*

P*rab-3* and fused to the last exon (blue) of *scyl-1* followed by 5 kb downstream genomic sequence. The red bars indicate the inverted repeat sequences that form the double-stranded RNA in the hair-pin structure (**B**). *wp1923* contains the intact genomic sequence of *scyl-1* 3'-UTR, whereas *wp1924* differs from it in an A-to-G conversion mimicking the ADR-1-dependent editing. (**E**) Effects of the A-to-G conversion on GFP reporter expression. Shown are fluorescent and corresponding DIC images of transgenic worms harboring either *wp1923* or *wp1924*. GFP expression in the head and ventral nerve cord (VNC) was observed only in worms harboring *wp1924*. The diffused signal below the VNC in fluorescent images was from auto-fluorescence of the intestine (Int). Scale bar = 20 μm. (**F**) GFP expression from *wp1924* was decreased in *adr-1(zw96)* compared with that in wild type. Scale bar = 20 μm. (**G**) Statistical comparison of GFP intensity in the VNC between *wild type* (n = 16) and *adr-1(zw96)* (n = 19). (***p<0.001, unpaired *t*-test).

The online version of this article includes the following source data and figure supplement(s) for figure 10:

**Source data 1.** Raw data and numerical values for data plotted in *Figure 10*.
**Figure supplement 1.** Expression of P*rab–3::GFP::scyl-1* 3'-UTR (A–to–G) was greatly decreased in isolated mutants.

---

Our results demonstrate that RNA editing at a single site in the 3'-UTR could have a profound effect on gene expression. The A-to-I conversion at the ADR-1-regulated editing site increases base pairing in the putative double-stranded structure of *scyl-1* 3'-UTR (*Figure 10B*). Increased base paring in a double-stranded RNA generally facilitates RNA degradation. It is therefore intriguing how such an editing event may cause increased gene expression. One

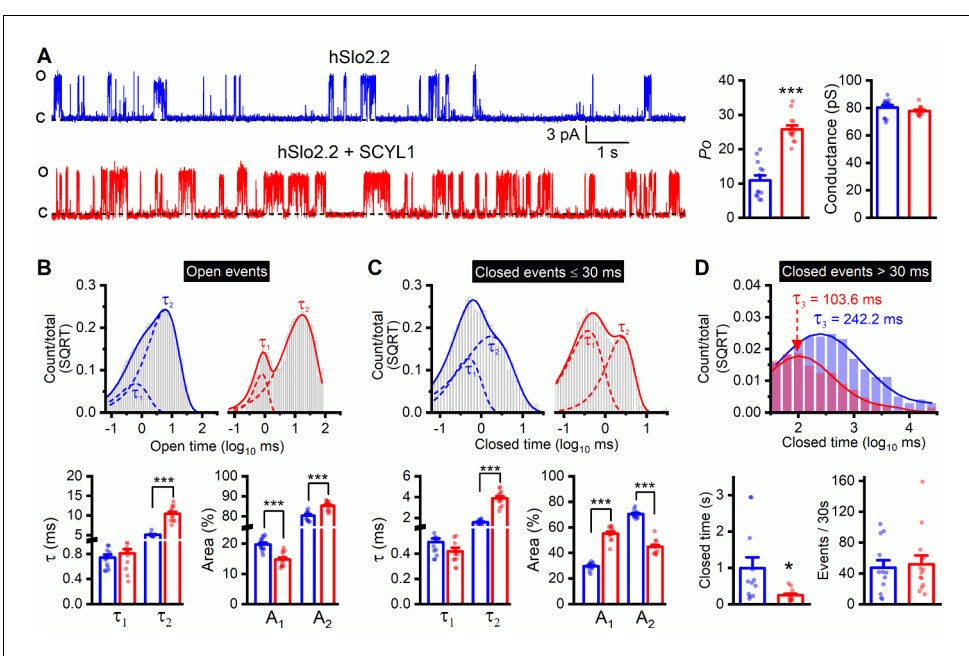

**Figure 11.** Single-channel open probability ($P_o$) of human Slo2.2/Slack is augmented by SCYL1 in *Xenopus* oocyte expression system. (**A**) Representative traces of single-channel currents from inside-out patches and comparisons of $P_o$ and single-channel amplitude between patches with and without mouse SCYL1. (**B and C**) Dwell time histograms and statistical comparisons of open and closed (≤30 ms in duration) events. The histograms were constructed and the τ values were quantified as described in *Figure 8* legend. (**D**) Dwell time histograms and statistical comparisons of the long closed events. Closed events that were >30 ms in duration of all recordings in each group were pooled together to construct the dwell time histogram. The average duration and frequency of these events were compared between the two groups. Each dot represents the mean value of one recording. Sample sizes were 13 in both groups. All values are shown as mean ± SE. The asterisks indicate a significant difference compared between the indicated groups (*p<0.05, ***p<0.001, unpaired *t*-test).

The online version of this article includes the following source data for figure 11:

**Source data 1.** Raw data and numerical values for data plotted in *Figure 11*.

possibility is that editing at this site helps maintain mRNA stability through recruiting other regulatory proteins to the 3'-UTR. The isolation of mutants exhibiting diminished GFP expression from a reporter construct containing the edited *scyl-1* 3'-UTR but not the *unc-10* 3'-UTR (*Figure 10—figure supplement 1*) indicates potential existence of such regulatory proteins. In combination with the observation that GFP signal was undetectable in transgenic worms expressing a reporter construct containing the non-edited *scyl-1* 3'-UTR, this result suggests that RNA editing at the ADR-1-dependent site in *scyl-1* 3'-UTR is likely important to recruiting the putative regulatory proteins to increase mRNA stability.

*scyl-1* has four different transcripts with identical 5'-UTR and coding sequence but different 3'-UTRs of variable lengths (ranging from 167 to 1862 nucleotides) and sequences (www.wormbase.org). The ADR-1-dependent editing site exists only in the transcript with the longest 3'-UTR. The presence of an ADR-1-regulated editing site in this but not the other transcripts suggests that ADR-1 may regulate *scyl-1* expression in a cell-specific manner depending on where the specific transcript is expressed. Interestingly, the 3'-UTR of human *SCYL1* transcripts (NM_020680.4) also has a high probability of forming hair-pin structures based on software prediction (https://rna.urmc.rochester.edu/RNAstructureWeb/Servers). It remains to be determined whether human *SCYL1* transcripts are also edited in the 3'-UTR, and if so, whether the editing regulates their expression.

SCYL1 proteins are evolutionarily conserved proteins that share an N-terminal pseudokinase domain (*Manning et al., 2002*; *Pelletier, 2016*). Results of previous studies with cultured cells suggest that SCYL1 may regulate intracellular trafficking between the Golgi apparatus and the ER (*Burman et al., 2008*; *Burman et al., 2010*), and facilitate nuclear tRNA export by acting at the nuclear pore complex (*Chafe and Mangroo, 2010*). Mutations of SCYL1 in humans are associated with a variety of disorders, including neurodegeneration, intellectual disabilities, and liver failure (*Lenz et al., 2018*; *Li et al., 2019*; *Schmidt et al., 2015*; *Shohet et al., 2019*; *Spagnoli et al., 2019*). Mice with SCYL1 deficiency develop an early onset and progressive neurodegenerative disorder (*Pelletier et al., 2012*). However, it is unclear whether the documented mutant phenotypes of SCYL1 are related to its roles in intracellular trafficking and nuclear tRNA export (*Pelletier, 2016*). This study brings into view a new potential mechanism for SCYL1 mutation-associated disorders: impairing Slo2 channel function. What might be the molecular mechanism through which SCYL-1 enhances SLO-2 activity? Since SCYL-1 physically associates with SLO-2, and enhances SLO-2 single-channel $P_o$ by altering the open and closed states, it likely regulates channel function either directly or through a closely associated protein. Although the exact mechanism remains to be determined, the similar effects of SCYL-1 and SCYL1 on Slo2 channels suggest that they likely play an important role in Slo2 physiological functions across animal species.

The expression patterns of *scyl-1* and *slo-2* largely do not overlap. Although they are coexpressed in ventral cord motor neurons, most other neurons expressing *slo-2* do not express *scyl-1*, suggesting that the regulatory effect of SCYL-1 on SLO-2 is cell- and tissue-specific. The expression of *scyl-1* in cells not expressing *slo-2* suggests that SCYL-1 physiological functions are not limited to regulating SLO-2. In mouse, SCYL1 and Slo2.2 are both expressed in the hippocampus and cerebellum but their expression patterns do not completely overlap (*Joiner et al., 1998*; *Schmidt et al., 2007*). Conceivably, the regulation of hSlo2.2 by SCYL1 might also occur in a cell- and tissue-specific manner, and SCYL1 likely performs other physiological functions. The latter possibility is supported by the pleiotropic phenotypes observed in patients and mice with SCYL1 mutations (*Lenz et al., 2018*; *Li et al., 2019*; *Pelletier et al., 2012*; *Schmidt et al., 2015*; *Shohet et al., 2019*; *Spagnoli et al., 2019*).

In summary, this study demonstrates that ADAR-mediated RNA editing controls the expression of SCYL-1, which interacts with SLO-2 to allow SLO-2 perform its physiological functions. Moreover, this study shows that this regulatory mechanism is conserved with mammalian SCYL1 and Slo2. Our findings reveal a new molecular mechanism of Slo2 channel regulation, and provide the bases for investigating how physiological functions of human Slo2 are regulated by SCYL1, and whether the neurodegeneration and intellectual disability phenotypes of SCYL1 mutations are related to Slo2 channel dysfunction.

# Materials and methods

## Key resources table

| Reagent type (species) or resource | Designation | Source or reference | Identifiers | Additional information |
|---|---|---|---|---|
| Strain, strain background (*C. elegans*) | N2 | Caenorhabditis Genetics Center | RRID:WB-STRAIN:WBStrain00000001 | Laboratory reference strain (wild type). |
| Strain, strain background (*C. elegans*) | LY101 | Caenorhabditis Genetics Center | RRID:WB-STRAIN:WBStrain00026423 | Genotype: *slo-2(nf101)*. |
| Strain, strain background (*C. elegans*) | BB3 | Caenorhabditis Genetics Center | RRID:WB-STRAIN:WBStrain00000435 | Genotype: *adr-2(gv42)*. |
| Strain, strain background (*C. elegans*) | ZW860 | This paper | | Genotype: *zwIs 139[Pslo-1::slo-2(gf)(wp1311), Pmyo-2::YFP(wp214)]*. |
| Strain, strain background (*C. elegans*) | ZW876 | This paper | | Genotype: *zwIs 139[Pslo-1::slo-2(gf) (wp1311), Pmyo-2::YFP (wp214)]; adr-1(zw80)*. |
| Strain, strain background (*C. elegans*) | ZW877 | This paper | | Genotype: *zwIs 139[Pslo-1::slo-2(gf) (wp1311), Pmyo-2::YFP (wp214)]; adr-1(zw81)*. |
| Strain, strain background (*C. elegans*) | ZW983 | This paper | | Genotype: *zwIs 139[Pslo-1::slo-2(gf) (wp1311), Pmyo-2::YFP (wp214)]; adr-2(gv42)*. |
| Strain, strain background (*C. elegans*) | ZW1049 | This paper | | Genotype: *zw Ex221[Prab-3::slo-2::GFP]*. |
| Strain, strain background (*C. elegans*) | ZW1388 | This paper | | Genotype: *zwEx260[Prab-3::His-58::mStrawberry (p1749), Prab-3::adr-1::GFP(p1374)]*. |
| Strain, strain background (*C. elegans*) | ZW1394 | This paper | | Genotype: *adr-1(zw96)*. |
| Strain, strain background (*C. elegans*) | ZW1401 | This paper | | Genotype: *zwEx261[Padr-1::GFP(wp1872), lin-15(+)]; lin-15(n765)*. |
| Strain, strain background (*C. elegans*) | ZW1407 | This paper | | Genotype: *zwIs139[Pslo-1::slo-2(gf)(wp1311), Pmyo-2::YFP(wp214)]; adr-1(zw96)*. |
| Strain, strain background (*C. elegans*) | ZW1408 | This paper | | Genotype: *zwIs139[Pslo-1::slo-2(gf)(wp1311), Pmyo-2::YFP(wp214)]; zwEx262[Prab-3::adr-1::GFP(p1374);Pmyo-2:: mStrawberry (wp1613)]; adr-1(zw96)*. |
| Strain, strain background (*C. elegans*) | ZW1409 | This paper | | Genotype: *scyl-1(zw99)*. ZW1410: *slo-2(nf101); scyl-1(zw99)*. |
| Strain, strain background (*C. elegans*) | ZW1415 | This paper | | Genotype:: *zwEx221[Prab-3::slo-2::GFP]; scyl-1(zw99)*. |

*Continued on next page*

*Continued*

| Reagent type (species) or resource | Designation | Source or reference | Identifiers | Additional information |
|---|---|---|---|---|
| Strain, strain background (*C. elegans*) | ZW1416 | This paper | | Genotype: *zwEx247[Pslo-2::mStrawberry(wp1776), lin-15(+)]; zwEx263[Pscyl-1::GFP(wp1901+wp1902), lin-15(+)]; lin-15(n765).* |
| Strain, strain background (*C. elegans*) | ZW1417 | This paper | | Genotype: *zwEx264[Prab-3::scyl-1(wp1912), Pmyo-2::mStrawberry (wp1613)]; scyl-1(zw99).* |
| Strain, strain background (*C. elegans*) | ZW1418 | This paper | | Genotype: *zwEx247[Pslo-2::mStrawberry(wp1776), lin-15(+)]; zwEx261[Padr-1::GFP(wp1872), lin-15 (+)]; lin-15(n765).* |
| Strain, strain background (*C. elegans*) | ZW1419 | This paper | | Genotype: *zwEx265[Prab-3::GFP::scyl-1 3-UTR (wp1923), lin-15(+)]; lin-15(n765).* |
| Strain, strain background (*C. elegans*) | ZW1420 | This paper | | Genotype: *zwEx266[Prab-3::GFP::scyl-1 3'-UTR(A-to-G) (wp1924), lin-15(+)]; lin-15(n765).* |
| Strain, strain background (*C. elegans*) | ZW1428 | This paper | | Genotype: *slo-2 (nf101); adr-1(zw96).* |
| Strain, strain background (*C. elegans*) | ZW1505 | This paper | | Genotype: *zwEx273[Prab-3::scyl-1::YFPc(wp1952), Prab-3::slo-2::YFPa (wp1783), lin-15(+)]; lin-15(n765).* |
| Strain, strain background (*C. elegans*) | ZW1506 | This paper | | Genotype: *zwEx274[Prab-3::scyl-1::YFPc(wp1952), Prab-3::slo-2N::YFPa (wp1784), lin-15(+)]; lin-15(n765).* |
| Strain, strain background (*C. elegans*) | ZW1507 | This paper | | Genotype: *zwEx275[Prab-3::scyl-1::YFPc(wp1952), Prab-3::slo-2C::YFPa (wp1785), lin-15(+)]; lin-15(n765).* |
| Strain, strain background (*C. elegans*) | ZW1537 | This paper | | Genotype: *adr-1(zw96);scyl-1(zw99).* |
| Strain, strain background (*C. elegans*) | ZW1538 | This paper | | Genotype: *zwEx280[Prab-3::scyl-1::HA(wp1998), lin-15(+)]; lin-15(n765).* |
| Strain, strain background (*C. elegans*) | ZW1539 | This paper | | Genotype: *zwEx281[Prab-3::scyl-1::HA(wp1998), Prab-3::slo-2::GFP (wp1318), lin-15(+)]; lin-15(n765).* |
| Strain, strain background (*C. elegans*) | ZW1540 | This paper | | Genotype: *zwEx282[Prab-3::scyl-1::HA(wp1998), Prab-3::slo-2N::GFP (wp1999), lin-15(+)]; lin-15(n765).* |

*Continued on next page*

Continued

| Reagent type (species) or resource | Designation | Source or reference | Identifiers | Additional information |
|---|---|---|---|---|
| Strain, strain background (*C. elegans*) | ZW1541 | This paper | | Genotype: *zwEx283[Prab-3::scyl-1::HA(wp1998), Prab-3::slo-2C::GFP (wp2000), lin-15(+)]; lin-15(n765).* |
| Strain, strain background (*C. elegans*) | ZW1544 | This paper | | Genotype: *zwIs146[Prab-3::GFP::scyl-1 3'-UTR (A-to-G)(wp1924)].* |
| Strain, strain background (*C. elegans*) | ZW1545 | This paper | | Genotype: *zwIs146[Prab-3::GFP::scyl-1 3'-UTR (A-to-G)(wp1924)]; adr-1(zw96).* |
| Strain, strain background (*C. elegans*) | ZW1549 | This paper | | Genotype: *zwIs146[Prab-3::GFP::scyl-1 3'-UTR (A-to-G)(wp1924)]; zw103.* |
| Strain, strain background (*C. elegans*) | ZW1552 | This paper | | Genotype: *zwIs146[Prab-3::GFP::scyl-1 3'UTR (A-to-G)(wp1924)]; zw105.* |
| Strain, strain background (*C. elegans*) | ZW1562 | This paper | | Genotype: *zwEx284[Prab-3::GFP::unc-10 3'-UTR(wp70)].* |
| Strain, strain background (*C. elegans*) | ZW1563 | This paper | | Genotype: *zwEx284[Prab-3::GFP::unc-10 3'-UTR (wp70)]; zw103.* |
| Strain, strain background (*C. elegans*) | ZW1564 | This paper | | Genotype: *zwEx284[Prab-3::GFP::unc-10 3'-UTR (wp70)]; zw105.* |
| Antibody | Mouse monoclonal anti-HA | Santa Cruz Biotechnology | Cat# sc-7392, RRID:AB_627809 | WB: 1:500 |
| Antibody | Mouse monoclonal anti-GFP | Santa Cruz Biotechnology | Cat# sc-9996, RRID:AB_627695 | WB: 1:500 |
| Antibody | Donkey anti-Mouse IgG | Thermo Fisher Scientific | Cat# A16011, RRID:AB_2534685 | WB: 1:10000 |
| Antibody | GFP-Trap_MA | ChromoTek | Cat# gtma-20, RRID:AB_2631358 | |
| Commercial assay or kit | ECL Substrate | Bio-Rad | Cat# 1705060 | |
| Commercial assay or kit | mMESSAGE mMACHINE | Ambion | Cat# AM1348 | |
| Software, algorithm | Photoshop CS5 | Adobe | RRID:SCR_014199 | https://www.adobe.com/products/photoshop.html |
| Software, algorithm | Origin | OriginLab | RRID:SCR_014212 | http://www.originlab.com/index.aspx?go=PRODUCTS/Origin |
| Software, algorithm | ImageJ | NIH | RRID:SCR_003070 | https://imagej.nih.gov/ij/ |
| Software, algorithm | pClamp | Molecular Devices | RRID:SCR_011323 | http://www.moleculardevices.com/products/software/pclamp.html |
| Software, algorithm | MATLAB | MathWorks | RRID:SCR_001622 | http://www.mathworks.com/products/matlab/ |
| Software, algorithm | TopHat | PMID:23618408 | RRID:SCR_013035 | http://ccb.jhu.edu/software/tophat/index.shtml |

*Continued on next page*

*Continued*

| Reagent type (species) or resource | Designation | Source or reference | Identifiers | Additional information |
|---|---|---|---|---|
| Software, algorithm | Trim Galore | Babraham Bioinformatics | RRID:SCR_011847 | http://www. bioinformatics. babraham.ac.uk/ projects/trim_galore/ |
| Software, algorithm | Track-A-Worm | PMID:23922769 | RRID:SCR_018299 | https://health.uconn. edu/worm-lab/ track-a-worm/ |

## *C. elegans* culture and strains

*C. elegans* hermaphrodites were grown on nematode growth medium (NGM) plates spotted with a layer of OP50 *Escherichia coli* at 22℃ inside an environmental chamber. All the strains used in this study are listed in the Key Resource Table (plasmids used in making the transgenic strains are indicated by numbers with a 'wp' prefix).

## Mutant screening and mapping

An integrated transgenic strain expressing P*slo-1*::SLO-2(*gf*) and P*myo-2*::YFP (transgenic marker) in the wild-type genetic background was used for mutant screen. L4-stage *slo-2(gf)* worms were treated with the chemical mutagen ethyl methanesulfonate (50 mM) for 4 hr at room temperature. F2 progeny from the mutagenized worms were screened under stereomicroscope for animals that moved better than the original *slo-2(gf)* worms. 17 suppressors were isolated in the screen and were subjected to whole-genome sequencing. Analysis of the whole-genome sequencing data showed that 2 mutants have mutations in the *adr-1* gene (www.wormbase.com). Identification of *adr-1* mutants was confirmed by the recovery of the sluggish phenotype when a wild-type cDNA of *adr-1* under the control of P*rab-3* was expressed in *slo-2(gf);adr-1(zw81)* double mutants.

## Generation of *adr-1* and *scyl-1* knockout mutants

The CRISPR/Cas9 approach (*Dickinson et al., 2013*) was used to create *adr-1* and *scyl-1* knockouts. The guide RNA sequences for *adr-1* and *scyl-1* are 5'- CCAGTTTTCGAAGCTTCGG and 5'- GAG-GAGATTGGAAAATTGG, which were inserted into pDD162 (P*eft-3*::Cas9 + Empty sgRNA; Addgene #47549), respectively. The resultant plasmids (wp1645 for *adr-1* and wp1887 for *scyl-1*) were injected into wild type worms, respectively, along with a repair primer (5'-GAGAAGTATTCACCAG TTTTCGAAGCTTAATGAGTTCCAAAAGATCCAGAGATTCCCGAA for *adr-1*, and 5'-TTGTAA-CAGCCGGAGGAGATTGGAAAATCTAGCTGGTGGACTTCATTTGGTCACTGGATT for *scyl-1*) and P*myo-2*::mStrawberry (wp1613) as the transgenic marker. The *adr-1* knockout worms were identified by PCR using primers 5'-TCACCAGTTTTCGAAGCTTAATGA (forward) and 5'-TCTTCTGCTGGC TCACATTCA (reverse). The *scyl-1* knockout worms were identified by PCR using primers 5'-CCGAAGTCCCAATTCCCAT (forward) and 5'- CCAAATGAAGTCCACCAGCTAG (reverse). The knockout worms were confirmed by Sanger sequencing.

## Analysis of expression pattern and subcellular localization

The expression pattern of *adr-1* was assessed by expressing GFP under the control of 1.8 kb *adr-1* promoter (P*adr-1*::GFP, wp1872). Primers for cloning P*adr-1* are 5'- TAAGGTACCAAGGACACG TTGCATATGAAT (forward) and 5'- TTTACCGGTTGGCTGACATATTGTGGGA (reverse). Subcellular localization of ADR-1 was determined by fusing GFP to its carboxyl terminus and expressing the fusion protein under the control of P*rab-3* (P*rab-3*::adr-1::GFP, wp1374). Primers for cloning *adr-1* cDNA are 5'- AAAGCGGCCGCATGGATCAAAATCCTAACTACAA (forward) and 5'- TTTACCGG TCCATCGAAAGCAGCAAGAGTGAAG (reverse). A plasmid (wp1749) harboring P*rab-3*::his-58:: mStrawberry serves as a nucleus marker. The expression pattern of *scyl-1* was assessed by an in vivo recombination approach. Specifically, a 0.5 kb fragment immediately upstream of *scyl-1* initiation site was cloned and fused to GFP using the primers 5'- AATCTGCAGCATCGGCACGAGAAGTACA (forward) and 5'- TTAGGATCCCTAAAAGTGATCGAAATTTA (reverse). The resultant plasmid (P*scyl-1*::GFP, wp1902) was linearized and co-injected with a linearized (fosmid WRM068bA03), which

contains 32 kb of *scyl-1* upstream sequence and part of its coding region, into the *lin-15(n765)* strain along with a *lin-15* rescue plasmid to serve as a transformation marker. To assay the effect of the identified adenosine site at the 3'UTR of *scyl-1* on gene expression, a 5.1 kb genomic DNA fragment covering part of the *scyl-1* last exon and subsequent sequence was cloned and fused in-frame to GFP using the primers 5'- AATGCTAGCATGCAGGCTAGAAATGAAGCTCG (forward) and 5'- TATGGGCCCGAAATCAGCATCTTTGACGAA (reverse). To mimic the A-to-I editing at the identified specific site, a second plasmid was made by mutating the specific adenosine to guanosine in the above plasmid. The two resultant plasmids were injected into *lin-15(n765)*, respectively, with a *lin-15* rescue plasmid as the transgenic marker. Images of transgenic worms were taken with a digital CMOS camera (Hamamatsu, C11440-22CU) mounted on a Nikon TE2000-U inverted microscope equipped with EGFP/FITC and mCherry/Texas Red filter sets (49002 and 49008, Chroma Technology Corporation, Rockingham, VT, USA).

## Behavioral assay

Locomotion velocity was determined using an automated locomotion tracking system as described previously (*Wang and Wang, 2013*). Briefly, a single adult hermaphrodite was transferred to an NGM plate without food. After allowing ~30 s for recovery from the transfer, snapshots of the worm were taken at 15 frames per second for 30 s using a IMAGINGSOURCE camera (DMK37BUX273) mounted on a stereomicroscope (LEICA M165FC). The worm was constantly kept in the center of the view field with a motorized microscope stage (OptiScanTM ES111, Prior Scientific, Inc, Rockland, MA, USA). Both the camera and the motorized stage were controlled by a custom program (*Source code 1*) running in MATLAB (The MathWorks, Inc, Natick, MA).

## RNA-seq and data analysis

Total RNA was extracted from young adult-stage worms using TRIzol Reagent (Invitrogen) and treated with TURBO DNase (Ambion). RNA-seq was performed by Novogene Corp. Sacramento, CA.

Raw reads ware filtered using Trim Galore software (http://www.bioinformatics.babraham.ac.uk/projects/trim_galore/) to remove reads containing adapters or reads of low quality. The filtered reads were mapped to *C. elegans* genome (*ce*11) using TopHat2 (*Kim et al., 2013*). The gene expression level is estimated by counting the reads that map to exons.

## Bimolecular fluorescence complementation (BiFC) assay

BiFC assays were performed by coexpressing SLO-2 and SCYL-1 tagged with the amino and carboxyl terminal portions of YFP (YFPa and YFPc), respectively, in neurons under the control of *rab-3* promoter (P*rab-3*). To assay which portion of SLO-2 may interact with SCYL-1, the full-length, N-terminal, and C-terminal portion of SLO-2 were fused with YFPa, respectively. The resultant plasmids (*wp1783*, P*rab-3*::SLO-2::YFPa; *wp1784*, P*rab-3*::SLO-2N::YFPa, and *wp1785*, P*rab-3*::SLO-2C::YFPa) were coinjected with P*rab-3*::SCYL-1::YFPc (*wp1952*), respectively, into *lin-15(n765)* strain. A *lin-15* rescue plasmid was also coinjected to serve as a transformation marker. Epifluorescence of the transgenic worms was visualized and imaged as described above.

## Co-immunoprecipitation

Mixed stage transgenic worms expressing either SCYL-1::HA alone or SCYL-1::HA with one of the GFP fusions (full-length SLO-2, SLO-2 amino-terminal, and SLO-2 carboxyl-terminal) were homogenized in lysis buffer containing 150 mM NaCl, 0.5 mM EDTA, 0.5 % P40, and 10 mM Tris/Cl pH 7.5. Immunoprecipitation was performed with a GFP-Trap Magnetic Agarose Kit (gtmak-20, ChromoTek Inc) according to the manufacturer's manual. The immune complexes and the worm lysates were separated on 4–20% Novex Tris-Glysine gels (XP04202BOX, Thermo Fisher Scientific) and probed with HA or GFP antibodies (sc-7392, sc-9996, Santa Cruz Biotechnology, Inc).

## *C. elegans* electrophysiology

Adult hermaphrodites were used in all electrophysiological experiments. Worms were immobilized and dissected as described previously (*Liu et al., 2007*). Borosilicate glass pipettes were used as electrodes for recording whole-cell currents. Pipette tip resistance for recording muscle cell currents

was 3–5 MΩ whereas that for recording motor neuron currents was ~20 MΩ. The dissected worm preparation was treated briefly with collagenase and perfused with the extracellular solution for 5 to 10-fold of bath volume. Classical whole-cell configuration was obtained by applying a negative pressure to the recording pipette. Current- and voltage-clamp experiments were performed with a Multiclamp 700B amplifier (Molecular Devices, Sunnyvale, CA, USA) and the Clampex software (version 10, Molecular Devices). Data were sampled at a rate of 10 kHz after filtering at 2 kHz. Spontaneous membrane potential changes were recorded using the current-clamp technique without current injection. Motor neuron whole-cell outward currents were recorded by applying a series of voltage steps (−60 to +70 mV at 10 mV intervals, 600 ms pulse duration) from a holding potential of −60 mV. Spontaneous PSCs were recorded from body-wall muscle cells at a holding potential of −60 mV. Two bath solutions and three pipette solutions were used in electrophysiological experiments as specified in figure legends. Bath solution I contained (in mM) 140 NaCl, 5 KCl, 5 $CaCl_2$, 5 $MgCl_2$, 11 dextrose and 5 HEPES (pH 7.2). Bath solution II contained (in mM) 100 $K^+$ gluconate, 50 KCl, 1 $Mg^{2+}$ gluconate, 0.1 $Ca^{2+}$ gluconate and 10 HEPES (pH 7.2). Pipette solution I contained (in mM) 120 KCl, 20 KOH, 5 Tris, 0.25 $CaCl_2$, 4 $MgCl_2$, 36 sucrose, 5 EGTA, and 4 $Na_2ATP$ (pH 7.2). Pipette solution II differed from pipette solution I in that 120 KCl was substituted by $K^+$ gluconate. Pipette solution III contained (in mM) 150 $K^+$ gluconate, 1 $Mg^{2+}$ gluconate and 10 HEPES (pH 7.2).

### *Xenopus* oocytes expression and electrophysiology

A construct containing human *Slack* cDNA (pOX + *hSlo2.2*, a gift from Dr. Salkoff) was linearized with Pvu I. The mouse *Scyl1* cDNA was amplified from a construct (MR210762, Origene) and cloned into an existing vector downstream of the T3 promoter. The resultant plasmid (*wp*1982) was linearized with NgoM4. Capped cRNAs were synthesized using the mMessage mMachine Kit (Ambion). Approximately 50 nl cRNA of either *Slack* alone (0.5 ng/nl) or *Slack* (0.5 ng/nl) plus *Scyl1* (0.5 ng/nl) was injected into each oocyte using a Drummond Nanoject II injector (Drummond Scientific). Injected oocytes were incubated at 18˚C in ND96 medium (in mM): 96 NaCl, 2 KCl, 1.8 $CaCl_2$, 1 $MgCl_2$, 5 HEPES (pH 7.5). 2 to 3 days after cRNA injection, single channel recordings were made in inside-out patches with a Multiclamp 700B amplifier (Molecular Devices, Sunnyvale, CA, USA) and the Clampex software (version 10, Molecular Devices). Data were sampled at 10 kHz after filtering at 2 kHz. Bath solution contained (in mM) 60 NaCl, 40 KCl, 50 $K^+$ gluconate, 10 KOH, 5 EGTA, and 5 HEPES (pH 7.2). Pipette solution contained (in mM) 100 $K^+$ gluconate, 60 $Na^+$ gluconate, 2 $MgCl_2$, and 5 HEPES (pH 7.2).

### Data analyses for electrophysiology

Amplitudes of whole-cell currents in response to voltage steps were determined from the mean current during the last 100 ms of the 1200 ms voltage pulses using the Clampfit software. The duration and charge transfer of PSC bursts were quantified with Clampfit software (version 10, Molecular Devices) as previously described (*Liu et al., 2013*). The frequency of PSC bursts was counted manually. For single channel analysis, the QuB software (https://qub.mandelics.com/) was used to fit open and closed times to exponentials, and to quantify the τ values and relative areas of the fitted components, which were automatically determined by the software. Single-channel conductance was calculated by dividing the single-channel current amplitude (determined from a Gaussian fit to the amplitude histogram) by the holding voltage. The first 30 s recording of each experiment was used for such analyses. Statistical comparisons were made with Origin Pro 2019 (OriginLab Corporation, Northampton, MA) using either *ANOVA* or unpaired *t*-test as specified in figure legends. $p < 0.05$ is considered to be statistically significant. The sample size (*n*) equals the number of cells or membrane patches analyzed. All values are shown as mean ± SE and data graphing was done with Origin Pro 2019.

## Acknowledgements

This work was supported by National Institute of Health (R01GM113004 to BC, and 2R01MH085927 and 1R01NS109388 to Z-WW). We thank Dr. Laurence Salkoff for the human *Slo2.2/Slack* construct. Some strains were provided by the CGC, which is funded by NIH Office of Research Infrastructure Programs (P40 OD010440).

## Additional information

### Funding

| Funder | Grant reference number | Author |
|---|---|---|
| National Institute of General Medical Sciences | R01GM113004 | Bojun Chen |
| National Institute of Mental Health | 2R01MH085927 | Zhao-Wen Wang |
| National Institute of Neurological Disorders and Stroke | 1R01NS109388 | Zhao-Wen Wang |

The funders had no role in study design, data collection and interpretation, or the decision to submit the work for publication.

### Author contributions

Long-Gang Niu, Ping Liu, Data curation, Formal analysis, Investigation, Writing - review and editing; Zhao-Wen Wang, Conceptualization, Resources, Supervision, Funding acquisition, Writing - review and editing; Bojun Chen, Conceptualization, Supervision, Funding acquisition, Writing - original draft, Project administration, Writing - review and editing

### Author ORCIDs

Bojun Chen (iD) https://orcid.org/0000-0002-1141-9101

### Decision letter and Author response

Decision letter https://doi.org/10.7554/eLife.53986.sa1
Author response https://doi.org/10.7554/eLife.53986.sa2

## Additional files

### Supplementary files

- Source code 1. Track-A-Worm software.
- Transparent reporting form

### Data availability

All data generated or analysed during this study are included in the manuscript and supporting files. Source data files have been provided for Figures 1, 2, 4, 5, 7, 8, 10,and 11. Sequencing data have been deposited in GEO under accession code GSE141316.

The following dataset was generated:

| Author(s) | Year | Dataset title | Dataset URL | Database and Identifier |
|---|---|---|---|---|
| Niu L, Liu P, Wang Z, Chen B | 2019 | Slo2 potassium channel function depends on a SCYL1 protein | https://www.ncbi.nlm.nih.gov/geo/query/acc.cgi?acc=GSE141316 | NCBI Gene Expression Omnibus, GSE141316 |

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
