## [Decision Letter]

Thank you for submitting your article "Slo2 potassium channel function depends on a SCYL1 protein" for consideration by *eLife*. Your article has been reviewed by Richard Aldrich as the Senior Editor, a Reviewing Editor (Oliver Hobert), and two reviewers. The following individuals involved in review of your submission have agreed to reveal their identity: Leonard Kaczmarek (Reviewer #1); Thomas Boulin (Reviewer #2).

The reviewers have discussed the reviews with one another and the Reviewing Editor has drafted this decision to help you prepare a revised submission. As you will see in the reviews below, there is general agreement about the general interest and importance of the study. However, a set of important clarifications and experiments need to be conducted to make this paper become acceptable for publication in *eLife*.

In brief, the requested experiments are:

1) Independent validation of interactions (reviewer #1, comment 2)

2) Engineering the A>G mutation into the genome (reviewer #2)

3) Improved expression pattern analysis (reviewer #2)(the in vivo recombineering technique has by now been shown to be inadequate).

There are also a number of very important clarifications that are required (e.g. reviewer #1, comment 1)

*Reviewer #1:*

This manuscript presents some very interesting work demonstrating that Slo2 channels in C-elegans interact with SYCL^-^1 to alter open probability of these channels. The effects of SYCL^-^1 on the channels is, in turn, influenced by ADR-2, which is required for A-to-I RNA editing of a site in the 3' UTR of the scyl-1 gene. The work is provocative and points the way to future work to unravel how this potential interaction affects the biology of neurons in the nematode and in mammalian systems.

1) My major comment is on the work that extrapolates the nematode findings to the human Slack channel. The latter has previously been shows to differ from the C-elegans channel in several ways. This interaction with the human channel is an important aspect of the presented work. Examination of the data in Figure 11, however, suggest that the gating of the human channel is quite different from the Slo2 channel records shown in Figure 8. Specifically (if the time bars shown in Figure 11A are correct, the overall open probability of the human channel is dominated by the presence of very long closed states (lasting many seconds). Examination of the X-axis scale bars in Figure 11B and C suggest these may have been excluded from the analyses. The conclusions on the effects of SCYL1 on open probability stated in the last paragraph of the Results section may not be completely valid until these are taken into account.

2) The bimolecular fluorescence complementation assay presented does make a case that there may be a physical interaction of SCLY-1 with SLO-2 but is not completely definitive. It would be good to have some other indicator of physical interaction to support this claim. A more conventional coimmunoprecipitation experiments would be a good addition, one that could perhaps be carried out using the co-expression *Xenopus* oocyte heterologous expression system.

*Reviewer #2:*

In this study Niu et al., describe a striking and entirely unsuspected regulatory cascade that controls SLO-2/Slo2 potassium channel activity. They report the role of SYCL^-^1, a novel physical interactor of the SLO-2 potassium channel, and describe how syCl^-^1 expression is controled by ADAR-dependent RNA editing in the non-coding sequence this gene. After dissecting this mechanism in worms, they proceed to directly demonstrate the conservation of this novel regulatory interaction with the human SLO-2 ortholog. Given the importance of this class of potassium channel in health and disease, identifying this modulatory mechanism is a very important finding in my opinion.

Essentiaol revisions:

- Please clarify the functional relationship between adr-1 and adr-2. Indeed, in the Introduction the authors seem to indicate that ADAR proteins could compete ("altering the accessibility") for certain binding sites. In this model, wouldn't one expect that adr-1(lf) would "free" access to the scyl-1 site, which would be inconsistent with the similarity of adr-1 and adr-2 mutant phenotypes?

Do the authors think that ADR-1 promotes the recruitement of ADR-2 to the scyl-1 3'UTR?

- I was very intrigued by the results described in Figure 10, and specifically Figure 10D/E. The result is not what I had intuitively expected. Since it was performed in an adr-1/2 wild-type background as far as I could determine from the Materials and methods section, I would have thought that ADAR activity would have edited the wild-type sequence and I would have expected to see GFP in both cases.

I would have performed this experiment in an adr-1(lf) background. Have the authors attempted this experiment in this background? How many independent lines were tested with the wp1923 construct? Is expression seen anywhere else in the nervous system (outside de VNC) to make sure that GFP can indeed be expressed from this transgene?

- Following on this previous point, a direct way to demonstrate the functional importance of this editing site would be to generate the following genotype by engineering the A>G mutation by CRISPR/Cas9 gene editing:slo-2(gf); adr-1(0); scyl-1(A>G)

My prediction would be that the mutation in scyl-1 would restore the slo-2 gain-of-function locomotor impairement, by bypassing the adr-1 requirement. To me this experiment would very strongly support this new and exciting functional regulation and I would encourage the authors to perform this rather simple experiment.

- In wormbase, the annotation of the scyl-1 locus shows a rather sizable 3'UTR. I was curious whether the authors have any comments on that point, and whether this is a common feature of ADAR-edited 3'UTRs.

Does the human SCYL1 3'UTR have a similar hair-pin structure, which could suggest a similar regulation mechanism?

- I was a bit surprised about the strategy used to generate the scyl-1 expression pattern. Why was the in vivo recombination approach used? What happens when the 0.5kb promoter-GFP construct is injected alone?

The previous gene is only approx 2kb upstream and there is significant sequence conservation to C. remanei and C. briggsae DNA less than 1kb upstream of the ATG (see UCSC genome browser for example). Did the authors test such a 2kb promoter fragment?

- Figure 6: I'm not convinced that scyl-1::GFP labels vm1 or vm2 muscles based on this image. I could be wrong, but higher magnification images would need to be checked.

---

## [Author Response]

Reviewer #1:1) My major comment is on the work that extrapolates the nematode findings to the human Slack channel. The latter has previously been shows to differ from the C-elegans channel in several ways. This interaction with the human channel is an important aspect of the presented work. Examination of the data in Figure 11, however, suggest that the gating of the human channel is quite different from the Slo2 channel records shown in Figure 8. Specifically (if the time bars shown in Figure 11A are correct, the overall open probability of the human channel is dominated by the presence of very long closed states (lasting many seconds). Examination of the X-axis scale bars in Figure 11B and C suggest these may have been excluded from the analyses. The conclusions on the effects of SCYL1 on open probability stated in the last paragraph of the Results section may not be completely valid until these are taken into account.

We thank the reviewer for the comment. The open probability in the original Figure 11 was calculated from all the open and closed events. As the reviewer pointed out, events of the very long closed state had a major impact on the open probability. However, they had little contribution to the dwell time histograms in the original figure because these events account for a tiny (nearly negligible) portion of the total event counts and the majority of them fell outside of the *x*-axis displayed in the figure. In the revised manuscript, we added a new panel (panel D) to Figure 11 to show dwell time histograms of the very long closed events (>30 ms, chosen arbitrarily as the threshold) for both groups. Because these events were scarce, our approach used for comparing τ and A values in panels B and C of this figure and of Figure 8 could not be applied to them. Therefore, we compared the average duration and frequency of these very long closed events between hSlo2.2 and hSlo2.2+SCYL1 (Figure 11D, bottom). We also have replaced the traces in Figure 11A with longer and more representative ones. In addition, the dwell time histograms in panels B and C of both Figures 8 and Figure 11, which were based on a single sample recording trace in the original figures, have been revised to include events from all the recordings to be consistent with the histogram in the new panel D. In the revised manuscript, we describe the presence of the very long closed state and its impact on the open probability (–subsection “scyl-1 expression depends on RNA editing at a specific 3’-UTR site”). The legends of Figure 8 and Figure 11 have also been updated accordingly.

2) The bimolecular fluorescence complementation assay presented does make a case that there may be a physical interaction of SCLY-1 with SLO-2 but is not completely definitive. It would be good to have some other indicator of physical interaction to support this claim. A more conventional coimmunoprecipitation experiments would be a good addition, one that could perhaps be carried out using the co-expression Xenopus oocyte heterologous expression system.

We performed coimmunoprecipitation assays with transgenic worms expressing HA-tagged SCYL-1 and GFP-tagged SLO-2. Worms instead of a heterologous expression system were used for this experiment because SLO-2 is poorly expressed in both *Xenopus oocytes* and HEK293 cells even with a codon-optimized version (our unpublished observation). Our results of the co-IP assays were in agreement with those of the BiFC assays (Figure 9), which provides further evidence for physical interactions between SLO-2 and SCYL-1.

Reviewer #2:Comments1) Please clarify the functional relationship between adr-1 and adr-2. Indeed, in the Introducution the authors seem to indicate that ADAR proteins could compete ("altering the accessibility") for certain binding sites. In this model, wouldn't one expect that adr-1(lf) would "free" access to the scyl-1 site, which would be inconsistent with the similarity of adr-1 and adr-2 mutant phenotypes?Do the authors think that ADR-1 promotes the recruitement of ADR-2 to the scyl-1 3'UTR?

We thank the reviewer for the comment. Indeed, our description about the functional relationship between ADR-1 and ADR-2 in the Introduction was inaccurate. ADR-1 promotes RNA editing by binding to target mRNAs and recruiting ADR-2 to some specific editing sites. Our results with *adr-1(lf)* and *adr-2(lf)* mutants are compatible with this knowledge. In the revised manuscript, a sentence in the Introduction has been changed to “ADR-1 is catalytically inactive but can promote RNA editing by binding to selected target mRNA and tethering ADR-2 to RNA substrates (Ganem et al., 2019; Rajendren et al., 2018; Washburn et al., 2014).” (Introduction)

2) I was very intrigued by the results described in Figure 10, and specifically Figure 10D/E. The result is not what I had intuitively expected. Since it was performed in an adr-1/2 wild-type background as far as I could determine from the Materials and methods section, I would have thought that ADAR activity would have edited the wild-type sequence and I would have expected to see GFP in both cases.I would have performed this experiment in an adr-1(lf) background. Have the authors attempted this experiment in this background? How many independent lines were tested with the wp1923 construct? Is expression seen anywhere else in the nervous system (outside de VNC) to make sure that GFP can indeed be expressed from this transgene?

We obtained five *wp1923* lines and three *wp1924* lines. While GFP signal was observed in all the *wp1924* lines, no GFP signal was detected anywhere in any of the *wp1923* lines. We were also surprised by the lack of GFP expression in the *wp1923* lines. As suggested, we performed the same experiments as those in Figure 10E with the *adr-1(zw96)* mutant. Specifically, *adr-1(zw96)* mutant strains expressing the *wp1923* transgene were made by injecting the *wp1923* plasmid into the mutant whereas that expressing the *wp1924* transgene by first integrating the *wp1924* transgene into the wild-type genome and then crossing it into *adr-1(zw96)*. As expected, GFP signal was not detected in the *adr-1(zw96)* mutant worms harboring the *wp1923* transgene (not shown). Interestingly, GFP signal from the *wp1924* transgene was much weaker (by ~50%) in the mutant strain than the wild-type strain (Figure 10 F and G). Because *scyl-1* 3’-UTR had already been “edited” at the ADR-1-dependent editing site in the *wp1924* transgene, the difference in GFP signal between wild-type and *adr-1(zw96)* strains suggests that ADR-1 can also regulate *scyl-1* mRNA level through a post-editing effect, perhaps through interacting with some other proteins. In agreement with this possibility, we were able to isolate several mutants showing decreased GFP signal from the *wp1924* transgene in a pilot genetic screen (Figure 10—figure supplement 1).

3) Following on this previous point, a direct way to demonstrate the functional importance of this editing site would be to generate the following genotype by engineering the A>G mutation by CRISPR/Cas9 gene editing:slo-2(gf); adr-1(0); scyl-1(A>G)My prediction would be that the mutation in scyl-1 would restore the slo-2 gain-of-function locomotor impairement, by bypassing the adr-1 requirement. To me this experiment would very strongly support this new and exciting functional regulation and I would encourage the authors to perform this rather simple experiment.

As suggested by the reviewer, we tried to engineer the A>G mutation at the ADR-1-dependent editing site in *scyl-1* 3’-UTR in the worm genome by CRISPR/Cas9. Unfortunately, our attempt was unsuccessful, probably because this editing site is located within an inverted sequence (Figure 10D). We then tried to address the reviewer’s comment by expressing P*rab-3::scyl-1::unc-10 3’-UTR* in *slo-2(gf);adr-1(lf)* double mutant and wild type (as a control) in the hope that expression of this transgene in the double mutant would restore the *slo-2(gf)* sluggish locomotion by bypassing the ADR-1 requirement. However, expression of the transgene caused larval arrest in the double mutant and a sluggish phenotype in wildtype worms, which prevented us from performing further analysis. Finally, we built a *scyl-1(zw99);adr1(zw96)* double mutant, and recorded whole-cell currents from VA5 motor neuron. We observed similar VA5 outward currents between the double mutant and the *scyl-1(zw99)* single mutant (Figure 7A), suggesting that SCYL-1 and ADR-1 likely contribute to SLO-2 function through a common pathway. This result provides further evidence for the putative role of ADR-1 in regulating SCYL-1 expression.

4) – In wormbase, the annotation of the scyl-1 locus shows a rather sizable 3'UTR. I was curious whether the authors have any comments on that point, and whether this is a common feature of ADAR-edited 3'UTRs.Does the human SCYL1 3'UTR have a similar hair-pin structure, which could suggest a similar regulation mechanism?

In Wormbase, there are four different *scyl-1* transcripts that differ only in the 3’-UTR sequence and length (167 nt, 169 nt, 564 nt, and 1862 nt). The identified ADR-1-dependent editing site exists only in the variant with the longest 3’UTR, which we speculate could be a mechanism for cell-specific expression of *scyl-1*. It is difficult to comment on whether RNA editing by ADARs at 3’-UTRs is a common mechanism of regulating mRNA expression based on the rather limited literatures. The 3’-UTR of human SCYL1 transcripts (NM_020680.4) also has a high probability of forming hair-pin structures based on software prediction (https://rna.urmc.rochester.edu/RNAstructureWeb/Servers). It remains to be determined whether human SCYL1 transcripts are also edited at the 3’-UTR, and if so, whether the editing regulates SCYL1 transcript expression. We added these comments to the Discussion section in the revised manuscript –.

*5) I was a bit surprised about the strategy used to generate the scyl-1 expression pattern. Why was the* in vivo recombination approach used? What happens when the 0.5kb promoter-GFP construct is injected alone?The previous gene is only approx 2kb upstream and there is significant sequence conservation to C. remanei and C. briggsae DNA less than 1kb upstream of the ATG (see UCSC genome browser for example). Did the authors test such a 2kb promoter fragment?

We initially tried to examine *scyl-1* expression pattern using a 2-kb P*scyl-1* because, as the reviewer pointed out, another gene (*lap-2*) resides approximately 2 kb upstream of *scyl-1*. However, GFP signal was not detected in transgenic worms expressing the P*scyl-1(2 kb)::gfp* transcriptional fusion. We therefore resorted to the commonly used in vivo homologous recombination approach to include potential distant upstream regulatory elements in the P*scyl-1::gfp* transcriptional fusion. A description about the unsuccessful experiment with the 2-kb P*scyl-1* has been added to the manuscript (–subsection “ADR-1 regulates SLO-2 function through SCYL-1”). As suggested by the reviewer, we also created transgenic worms expressing only the P*scyl-1(0.5 kb)::gfp* transcriptional fusion but were unable to detect any GFP signal from them (not shown).

6) Figure 6: I'm not convinced that scyl-1::GFP labels vm1 or vm2 muscles based on this image. I could be wrong, but higher magnification images would need to be checked.

We thank the reviewer for pointing out our mistake. In Figure 6, P*scyl-1::GFP* labels uterine ventral cells rather than vulval muscle cells.